# Nuclear envelope-associated lipid droplets are enriched in cholesteryl esters and increase during inflammatory signaling

Ábel Szkalisity [1,2,7], Lauri Vanharanta [1,2,7], Hodaka Saito [1,2], Csaba Vörös [1,2,3], Shiqian Li [1,2], Antti Isomäki [4], Teemu Tomberg [5], Clare Strachan [5], Ilya Belevich [6], Eija Jokitalo [6] & Elina Ikonen [1,2✉]

## Abstract

Cholesteryl esters (CEs) and triacylglycerols (TAGs) are stored in lipid droplets (LDs), but their compartmentalisation is not well understood. Here, we established a hyperspectral stimulated Raman scattering microscopy system to identify and quantitatively assess CEs and TAGs in individual LDs of human cells. We found that nuclear envelope-associated lipid droplets (NE-LDs) were enriched in cholesteryl esters compared to lipid droplets in the cytoplasm. Correlative light-volume-electron microscopy revealed that NE-LDs projected towards the cytoplasm and associated with type II nuclear envelope (NE) invaginations. The nuclear envelope localization of sterol O-acyltransferase 1 (SOAT1) contributed to NE-LD generation, as trapping of SOAT1 to the NE further increased their number. Upon stimulation by the pro-inflammatory cytokine TNFα, the number of NE-LDs moderately increased. Moreover, TNFα-induced NF-κB nuclear translocation was fine-tuned by SOAT1: increased SOAT1 activity and NE-LDs associated with faster NF-κB translocation, whereas reduced SOAT1 activity and NE-LDs associated with slower NF-κB translocation. Our findings suggest that the NE is enriched in CEs and that cholesterol esterification can modulate nuclear translocation.

**Keywords** Label-Free Imaging; Lipid Droplet; Neutral Lipid; Nuclear Envelope
**Subject Categories** Immunology; Membranes & Trafficking; Organelles

## Introduction

Lipid droplets (LDs) are subcellular organelles that serve as cellular reservoirs of lipids and lipophilic compounds. They are comprised of a hydrophobic core filled with neutral lipids surrounded by a phospholipid monolayer coated with proteins. LDs are generated in the endoplasmic reticulum (ER) and form contact sites with various organelles, and can even appear in the nucleus (Schuldiner and Bohnert, 2017; Ohsaki et al, 2016; Zadoorian et al, 2023; Thiam and Ikonen, 2021; Mathiowetz and Olzmann, 2024). Understanding the multifunctional roles of LDs requires information about the chemical composition of LDs at distinct subcellular sites.

The major neutral lipids stored in LDs are triacylglycerols (TAGs) and cholesteryl esters (CEs), but LD contents vary considerably between cell types. Adipocytes represent professional cells for storing chemical energy in the form of TAGs, while steroidogenic cells store CEs as precursors for steroid hormone synthesis. The lipid composition of LDs affects the associated proteins, for instance various perilipins associate with LDs of different neutral lipid compositions (Hsieh et al, 2012). Moreover, while some cells, such as steroidogenic adrenal cells, were shown to store CEs and TAGs in separate LDs (Alfonso-García et al, 2015), other cells generate mixed LDs containing both CEs and TAGs, as demonstrated e.g., for macrophages (Bautista et al, 2014). Indeed, ongoing TAG synthesis can facilitate CE storage by generating TAG pre-clusters or "seeds" that catalyze the nucleation of CE LDs (Dumesnil and Vanharanta et al, 2023).

Yet, how CEs and TAGs are distributed between individual LDs is not well characterized. This has largely been due to the challenges in specifically detecting CEs and TAGs by spatially resolving techniques. Fluorescence microscopy provides high spatial resolution, but fluorescent labels alter chemical specificity, are often relatively big and hydrophilic compared to lipids, and may interfere with lipid metabolism. Developments in label-free imaging techniques, such as coherent anti-Stokes Raman scattering (CARS) microscopy and stimulated Raman scattering (SRS) microscopy, enable the study of lipids in cells and tissues with the spatial resolution of confocal microscopy (Hu et al, 2019). Moreover, they allow native TAGs and CEs to be chemically resolved based on their distinct vibrational spectra (Hislop et al, 2022; Xu et al, 2024).

In this study, we established a hyperspectral SRS microscopy regime to assess the CE and TAG composition of individual LDs of human cells in a quantitative manner. Our results demonstrate that this SRS-based spatially resolved neutral lipid quantification agrees

[1]Department of Anatomy and Stem Cells and Metabolism Research Program, Faculty of Medicine, University of Helsinki, 00014 Helsinki, Finland. [2]Minerva Foundation Institute for Medical Research, 00290 Helsinki, Finland. [3]Synthetic and Systems Biology Unit, Biological Research Centre (BRC), Hungarian Research Network (HUN-REN), 6726 Szeged, Hungary. [4]Biomedicum Imaging Unit, Department of Anatomy, Faculty of Medicine, University of Helsinki, 00290 Helsinki, Finland. [5]Division of Pharmaceutical Chemistry and Technology, Faculty of Pharmacy, University of Helsinki, 00014 Helsinki, Finland. [6]Electron Microscopy Unit, Institute of Biotechnology, Helsinki Institute of Life Science, University of Helsinki, Helsinki, Finland. [7]These authors contributed equally: Ábel Szkalisity, Lauri Vanharanta. ✉E-mail: elina.ikonen@helsinki.fi

well with conventional biochemical lipid analysis. Using this system, we identified differential neutral lipid compositions between LDs associated with the nuclear and cytoplasmic endoplasmic reticulum (ER) domains and found evidence that cholesterol esterification is physiologically relevant for controlling nuclear transport, e.g. in the context of initiating inflammatory signaling.

# Results

## SRS imaging and image analysis

The cellular SRS signal derives from a wide range of biomolecules. To measure the amount of a specific biomolecule, e.g., a neutral lipid, from this confounded signal, the signal of interest needs to be extracted by data analysis. To this end, we decided to decompose the cellular SRS signal to the linear combination of predefined reference spectra. We chose to obtain TAG and CE references from cellular LDs rather than from pure chemicals to better model cell-type specific mixtures of neutral lipids with different fatty acid compositions.

Initially, we used human epithelial carcinoma A431 cells employed extensively in our earlier LD-related studies (Salo et al, 2016, 2019; Prasanna et al, 2021; Dumesnil et al, 2023) to generate "reference cells" with predominantly TAG or CE containing LDs. We first starved the cells in lipoprotein-deficient serum (LPDS)-containing medium in the presence of diacylglycerol acyltransferase 1 and 2 (DGAT1&2) inhibitors for one day to deplete their existing lipid stores. We then loaded them for 2 h with either oleic acid (OA) in the presence of a sterol O-acyltransferase 1 (SOAT1) inhibitor or with cholesterol in the presence of DGAT1&2 inhibitors. High-performance thin-layer chromatography (TLC) analysis confirmed that the cells stored the expected neutral lipid classes, i.e., TAG in the case of OA and CE in the case of cholesterol loading (Fig. 1A).

We next acquired SRS images of such TAG or CE-containing cells in the wavenumber region characteristic of lipids. As this detects not only LDs but also lipid-rich membrane compartments (Fig. 1B), we segmented LDs from the images based on their morphology (Fig. 1C). Upon plotting the average spectrum of LDs for both conditions, we obtained spectra with the strongest peaks at wavenumbers 2857 and 2875 cm$^{-1}$, matching the previously reported peaks of TAG and CE Raman spectra (Fig. 1C). The former peak originated from the symmetric $CH_2$ stretching in the open fatty acid chains and the latter from the symmetric $CH_2$ stretching within the sterol ring (Czamara et al, 2015); but for improved lipid distinction, the entire spectral information was considered.

When acquiring the SRS spectra from cellular LDs, signals from cellular membranes overlapping with LDs are also collected. To reduce their spectral contribution to the TAG and CE references, we defined a cellular background spectrum measured from lipid-starved cells and subtracted it from the spectra of TAG and CE LDs. In addition, we defined a water background reference spectrum from regions devoid of cells (Fig EV1A,B).

The four references (i.e., TAG, CE, cellular, and water background) were then used to determine the neutral lipid composition of cells of interest. We utilized a non-negative least squares approximation to decompose the SRS signal from each pixel into the linear combination

of the references (Figs. 1D and EV1B). The weights from the linear combination for TAG and CE references were considered direct estimates of TAG and CE lipid amounts (Fig. EV1C; see Methods for details). For visualization, the weights were scaled and back-projected to the SRS images to create false-colored images of TAGs and CEs with magenta and green (Figs. 1D and EV1D,E). As a proof-of-principle, the decomposition of OA and cholesterol-only loaded cells showed mostly purely magenta and green droplets (Fig. 1E). Together, these data demonstrate that the linear decomposition based on predefined reference spectra can reliably distinguish TAG and CE LDs.

## Distribution of CEs and TAGs in cells loaded with cholesterol and OA

Next, we studied the composition of LDs in A431 cells loaded with both OA and cholesterol using SRS imaging. Such double loaded cells store both CE and TAG, and cholesterol loading together with OA considerably increases CE storage compared to loading with cholesterol alone, as we recently reported (Dumesnil and Vanharanta et al, 2023). To visualize the distribution of CEs and TAGs in the LDs of double loaded cells, we employed the reference based linear decomposition of signals described above.

This showed that most LDs in A431 cells contained both CEs and TAGs, as expected and that overall, the distribution of CEs and TAGs between differently sized LDs was relatively uniform (Fig. 1F). Interestingly, the distribution of CEs and TAGs inside individual LDs often appeared to be non-uniform, with CEs typically appearing as a shell around a TAG-rich core (Fig. 1G). This CE signal was missing from cells loaded with cholesterol and OA in the presence of a SOAT1 inhibitor (Fig. 1G), arguing for the specificity of the SRS CE signal. Indeed, such peripheral CE distribution in LDs agrees with cryo-electron tomography data (Mahamid et al, 2019; Rogers et al, 2022).

As SRS imaging is quantitative in nature (Freudiger et al, 2008), the estimates of CE and TAG amounts based on SRS imaging should, in principle, match with biochemical lipid analysis. We found that the SRS-based estimates of lipid amounts matched nicely with TLC analyses performed from parallel samples under the different lipid loading conditions (Fig. 1H). Of note, it is important to ensure that the SRS signal does not get saturated, as this alters the shape of the signal and, thereby, the estimation of lipid amounts.

The most notable difference between SRS and TLC-based quantitation was in the estimation of CE amount under conditions of cholesterol-only loading (Fig. 1H). Considering that CE is challenging to pack into LDs and that the membrane bilayer can hold substantial amounts of CE (Dumesnil and Vanharanta et al, 2023), we analyzed the whole-cell SRS signal from cholesterol-loaded cells (Fig. EV1E). This suggested that there is a substantial amount of CE outside LDs, presumably in the ER. Indeed, summing up this pool of CEs with the LD CE pool detected by SRS provided a better match with biochemically determined CE levels (Fig. EV1F).

To assess the SRS-based neutral lipid detection in a different setting, we turned to steroidogenic cells that prioritize LDs storing CEs as reservoirs for steroid hormone production. Analogously to A431 cells, we starved KGN cells deriving from a human ovarian granulosa tumor in an LPDS-containing medium and then loaded them with OA, cholesterol, or both lipids, followed by SRS imaging and signal decomposition. As for A431 cells, KGN cells loaded only

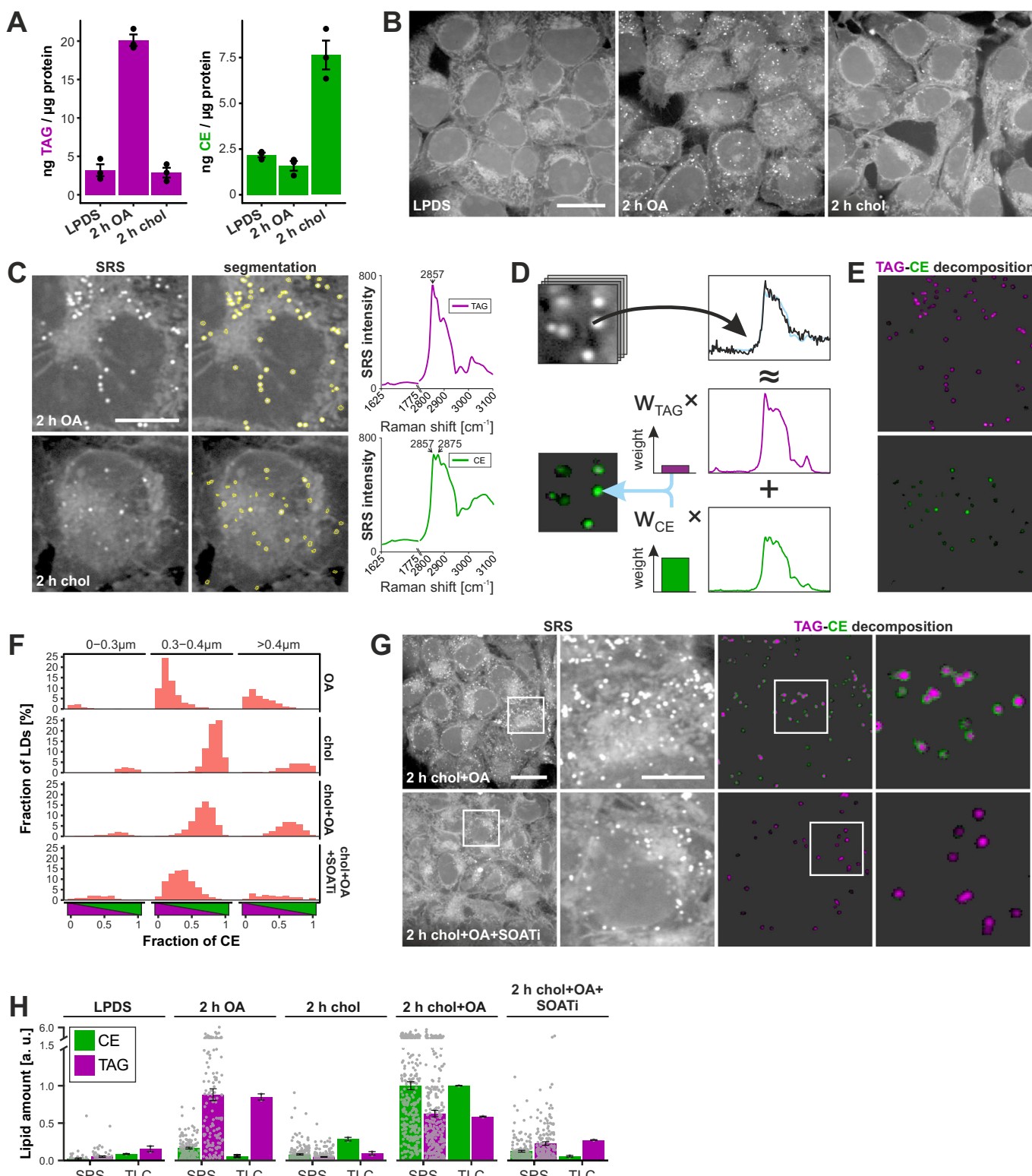

with cholesterol (in the presence of DGAT inhibitors) or OA (in the presence of SOAT inhibitor) were imaged to acquire reference spectra for CE and TAG. The results showed that while KGN cells can store TAG upon OA loading, they strongly favor CE deposition when fed with either cholesterol or both cholesterol and OA (Fig. 2A,B). Interestingly, OA addition, together with cholesterol, made the CE LDs substantially bigger than the CE-rich LDs forming upon cholesterol storage alone (Fig. 2A,C).

◄ **Figure 1.  Decomposition of hyperspectral SRS signal into neutral lipid classes.**

(A) TLC analysis of lipid-starved and cholesterol (chol) or oleic acid (OA) loaded A431 cells (mean ± SEM). (B) Representative SRS images of cells treated as in (A). (C) Exemplary SRS images showing LD segmentation with yellow contours. The average spectra of LDs from chol and OA-loaded cells are visualized in the 1625–1775 and 2800–3100 cm⁻¹ Raman shift regions with a step size 3 cm⁻¹. SRS intensity is measured as Stimulated Raman loss. (D) Illustration of spectrum decomposition into TAG and CE components. Each pixel's raw spectrum (black curve) is estimated (light blue curve) as a non-negative weighted sum of TAG, CE, and background reference spectra (see Fig. EV1B). The assigned weight is projected back to the image, indicating the amount of TAG and CE with magenta and green colors, respectively. (E) Result of the linear decomposition of the SRS images in (C) in the form of back-projected and color-coded weights. (F) Histogram of LDs with respect to their size and neutral lipid composition in A431 cells loaded for 2 h, as indicated. The fraction of CE is calculated as CE/(CE + TAG). LD sizes are binned based on their radius. (G) Representative SRS images and lipid decompositions of chol and OA-loaded A431 cells without and with SOAT inhibition (SOATi). (H) Comparison of CE and TAG quantification using SRS and duplicate TLC samples, prepared in parallel (mean ± SEM). All values normalized to the CE amount of chol+OA load in both TLC and SRS-based quantification. Cell numbers: 61, 132, 205, 268, 172 cells for LPDS (only H), OA, chol, chol+OA, chol+OA+SOATi in (F, H). Scale bars: 25 µm (B, G—overview), 10 µm (C, G—inset). Source data are available online for this figure.

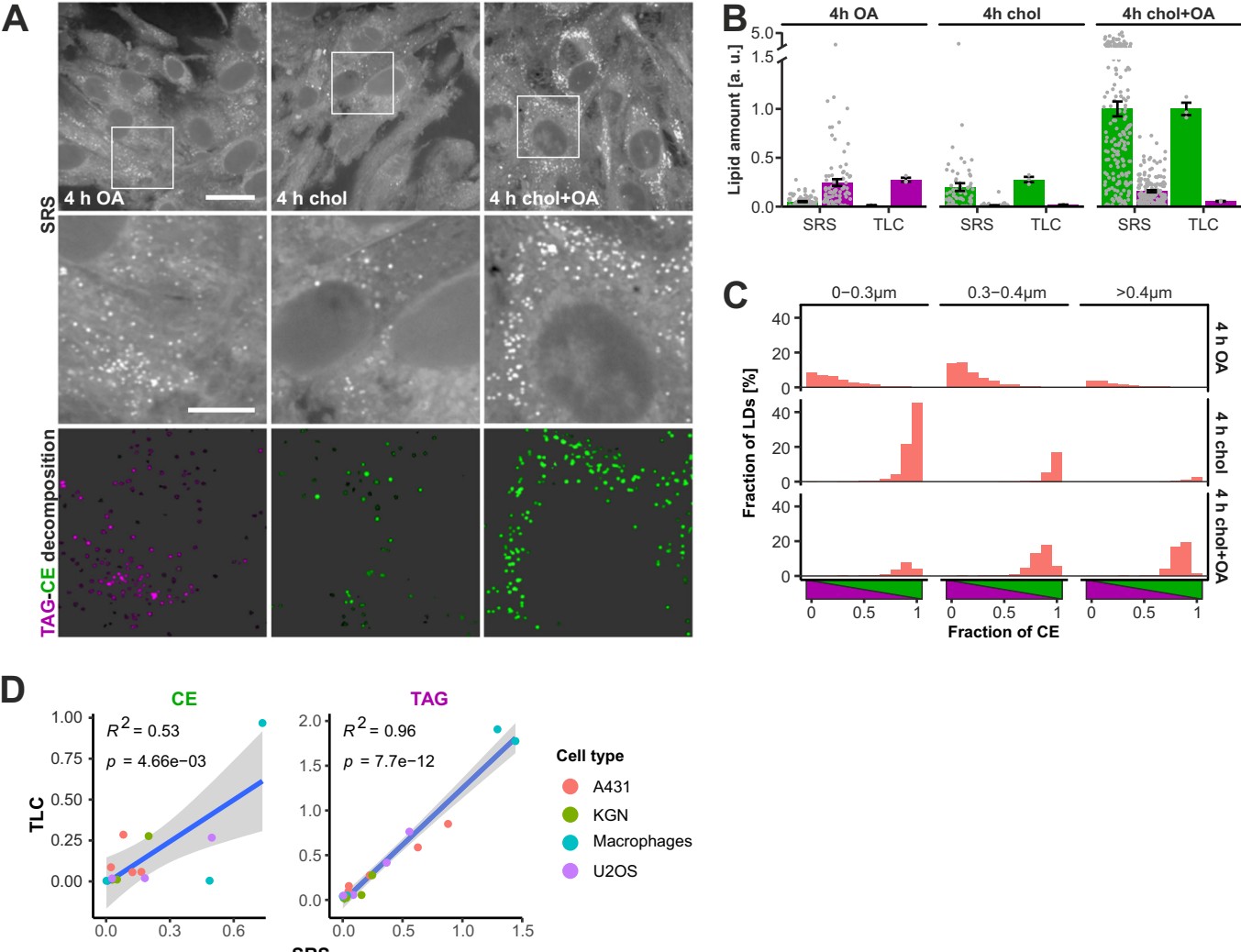

**Figure 2.  SRS analysis of neutral lipids in LDs of steroidogenic cells.**

(A) Representative SRS images and LD decomposition of KGN cells lipid loaded for 4 h as indicated. (B) Comparison of lipid quantifications with SRS and TLC as in Fig. 1H, triplicate samples for TLC with parallel samples for SRS imaging (mean ± SEM). (C) Histogram of LDs with respect to their size and neutral lipid composition in KGN cells lipid loaded for 4 h as indicated. (D) Correlation between SRS and TLC-based estimates of CE and TAG amounts in the cell types indicated. Pearson correlation coefficient, p value from F-test. Cell numbers: 87, 64, 174 cells for OA, chol, chol + OA in (B, C). Scale bars: 25 µm (A—overview), 10 µm (A—inset). Source data are available online for this figure.

Furthermore, biochemical lipid analysis performed in parallel showed a good match between the SRS and TLC-based lipid detection (Fig. 2B). Of note, in cholesterol-only loaded KGN cells, the CE SRS signal detected from LDs matched better with the whole-cell CE amount detected by TLC, suggesting that KGN cells may be more efficient than A431 cells in packaging CEs to LDs. Finally, additional analyses in cholesterol and OA-loaded U2OS cells and primary human monocyte-macrophages showed that SRS and TLC-based assessments of neutral lipid amounts correlated well (Fig. 2D).

## Nuclear envelope-associated lipid droplets are CE-enriched

We next investigated potential differences in CE and TAG storage between LDs more systematically. It is easy to distinguish the nuclear area in SRS images, hence our attention turned to LDs in the nuclear region vs. the rest of the cell. LDs in the nuclear region were readily visible in the SRS images as droplets surrounded by the darker nuclear area, verified by DAPI staining (Fig. 3A,B). Based on z stacks, such droplets often seemed to localize either above or below nuclei, and sometimes apparently associated with membrane structures intruding into the nucleus (Fig. 3B).

To better map the localization of LDs within the nuclear area, we used correlative light-volume-EM (CLvEM), specifically confocal light microscopy (LM) imaging followed by serial block face scanning electron microscopy (SBF-SEM). SBF-SEM is suitable to image entire volumes of cells with up to 30-nm longitudinal voxels. To compensate for the limited throughput of SBF-SEM, the CLvEM approach was used to ensure imaging of cells with LDs in the nuclear area (Fig. EV2A–C). Imaging of control and 2 h cholesterol+OA-loaded cells showed that LDs in close proximity to the NE were localized on the cytoplasmic side (Fig. 3C). Hence, we termed them nuclear envelope-associated LDs (NE-LDs) to distinguish them from other LDs exposed to the cytoplasm that we termed cytoplasmic LDs. From the completely imaged representative A431 cells (1 control and 2 loaded ones) no examples of LDs facing the nucleoplasm were found, although tubular and sheet-like nucleoplasmic reticulum (Malhas et al, 2011) invaginations projecting towards the nucleoplasm were present (Fig. 3C, arrowheads). NE-LDs were found in dents on the nucleus (both above and under), with most of them facing the adherent side of the cell (Fig. 3C, arrows).

The decomposed, false-colored SRS images of cholesterol and OA-loaded A431 cells revealed that NE-LDs were typically greener than cytoplasmic LDs, indicating a higher CE content (Fig. 3D). Indeed, when quantifying the CE enrichment of LDs as defined by their CE/(CE + TAG) ratio, we found that NE-LDs systematically contained a moderately but significantly higher fraction of CEs than cytoplasmic LDs (Fig. 3E). Remarkably, this was not unique for A431 cells, but was also observed in cholesterol and OA-loaded U2OS cells and primary human monocyte-derived macrophages (Fig. 3D,E). Thus, NE-LDs were found to be more CE enriched than cytoplasmic LDs in several cell types.

## Volume-EM analysis of NE-LDs in human macrophages

As NE-LDs were prevalent in the cholesterol and OA-treated, lipid-laden human primary macrophages, we also analyzed them using CLvEM, focusing on cells with multiple LDs in the nuclear area. Intriguingly, in cells enriched with NE-LDs, we observed NE

membrane invaginations of various extents ranging from deep finger-like membrane invaginations (Figs. 4A,B and EV3A) to complete tunnels formed by the NE and extending through the whole nucleus (Figs. 4C,D and EV3B). In some cases, the individual or clustered NE-LDs were found at the mouth of these tunnels (Fig. 4B,D), but tunnels filled with LDs were also observed (Figs. 4E,F and EV3C). Interestingly, the NE forming the tunnel through the nucleus and surrounding NE-LDs contained nuclear pores (Fig. 4G). Of note, such NE invaginations of various depths were also found in non-loaded macrophages (Appendix Fig. S1). Together, these data suggest that NE-LDs were often localized in the proximity of NE invaginations.

## Trapping of SOAT1 to the NE increases CE storage in NE-LDs

SOAT1 is the enzyme primarily responsible for esterifying cholesterol into CEs. Accordingly, stable overexpression of SOAT1-GFP in A431 cells resulted in increased CE storage (Fig. 5A), while unesterified cholesterol levels remained similar (26.1 ± 2.32 ng/µg protein in wildtype and 24.1 ± 1.41 ng/µg protein in SOAT1-GFP overexpressing A431 cells). We have earlier reported that a fraction of endogenous SOAT1 localizes to the NE in A431 cells (Lak et al, 2021). We, therefore, hypothesized that the NE localization of SOAT1 contributes to the formation of CE-rich LDs at the NE. To test this, we trapped the overexpressed SOAT1-GFP to the NE using a C-terminal fusion to KASH in combination with SUN overexpression (Sosa et al, 2012) (Fig. 5B). The trapped SOAT1-GFP was as active as the non-trapped enzyme, based on the levels of CEs produced upon cholesterol or cholesterol and OA loading (Fig. 5C). Interestingly, the amount of TAG was lower in the SOAT1-NE-trapped cells compared to the non-trapped SOAT1 expressing cells (Fig. 5C), probably because the CE and TAG synthesizing enzymes had lost their spatial coordination.

Combined fluorescence and SRS imaging showed that the NE trapping of SOAT1-GFP was highly efficient and significantly increased the number and fraction of NE-LDs under cholesterol and cholesterol+OA loading conditions (Fig. 5D–G). Consequently, the proportion of CE stored in NE-LDs was higher in the NE-trapped SOAT1 cells than in wildtype or SOAT1 overexpressing cells (Fig. 5H). We also noted that the nuclear shape became irregular in the SOAT1-NE-trapped cells (Fig. 5D). However, this was related to the tethering as it was independent of SOAT1 activity (Appendix Fig. S2). Together, these results argue that the NE localization of SOAT1 assists in the generation of CE-rich LDs in the NE.

## TNFα induces SOAT1 redistribution and CE-enriched LDs in the NE

We next searched for potential physiological cues that might increase CE-rich NE-LDs in cells. It is well-known that plasma membrane cholesterol can be mobilized by enzymatic hydrolysis of sphingomyelin, which reduces the fraction of sphingomyelin-complexed cholesterol (Slotte and Bierman, 1988) and increases cholesterol accessibility for transport to the ER and esterification by SOAT1 (Das et al, 2014). Neutral sphingomyelinase activity, in turn, is stimulated by various pro-inflammatory agents, such as tumor necrosis factor alpha (TNFα) and lipopolysaccharide

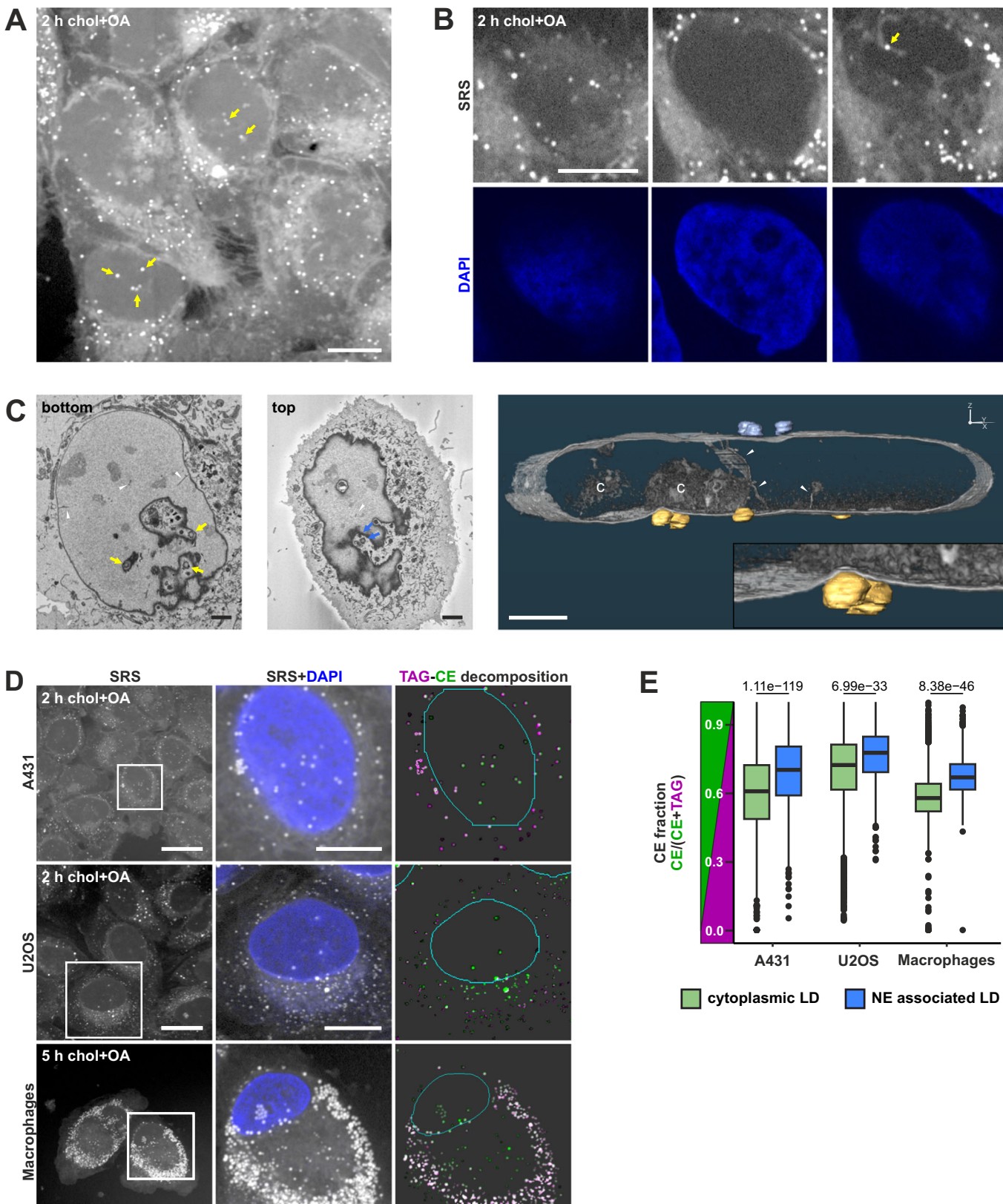

**Figure 3.   Nuclear envelope-associated LDs are enriched in CE compared to cytoplasmic LDs.**

(A) Examples of NE-LDs (yellow arrows) in an SRS image of A431 cells loaded with chol and OA for 2 h. (B) Z-stack slices from SRS and DAPI images of an A431 cell treated as in (A). The yellow arrow indicates an NE-LD colocalizing with an NE invagination. Distance of consecutive slices: 2.25 μm. (C) Block face images of a bottom and a top cross-section from SBF-SEM dataset of an A431 cell in complete medium showing the relationship of NE-LDs to the NE (arrows), and volume rendering of an NE cross-section with the models of the associated LDs (below the nucleus in yellow, above in light blue). 3D visualization reveals shallow dents in the NE adjacent to LDs (inset) and sheet-like nuclear reticulum invaginations (arrowheads) protruding into or across the nucleus; c denotes condensed chromatin. See Fig. EV2 for CLvEM workflow snapshots. (D) SRS imaging of NE-LDs in different cell types loaded with chol and OA as indicated. The nuclear area is indicated by DAPI and a cyan line in the insets. (E) Fraction of CE in the cytoplasmic and NE-LDs from each cell type in (D). p values from t-tests and boxplots according to ggplot (see Methods). LD numbers for cytoplasmic and NE-LDs: 5639 and 1538 for A431 from 204 cells; 5848 and 863 for U2OS from 183 cells, and 9366 and 522 for primary human macrophages from 39 cells. Scale bars: 10 μm (A, B, D—inset), 2 μm (C), 25 μm (D—overview). Source data are available online for this figure.

(Kim et al, 1991; Haimovitz-Friedman et al, 1997), increasing CE storage (Chatterjee, 1994).

As TNF-receptor 1 is ubiquitously expressed in most cells, we chose to test the effect of acute TNFα treatment on NE-LD formation (Fig. 6A). Indeed, when A431 cells were treated for 30 or 60 min with TNFα, the number of CE-enriched NE-LDs increased as assessed by SRS imaging (Fig. 6B–D). This effect was abrogated when SOAT1 inhibitor was added together with TNFα (Fig. EV4A–C). Together, these data suggest that TNFα treatment rapidly mobilizes endogenous plasma membrane cholesterol that moves to the ER for esterification and becomes deposited in NE-LDs.

To investigate if TNFα affects the localization of SOAT1, we SNAP-tagged endogenous SOAT1 in A431 cells and monitored its distribution before and after TNFα treatment. This revealed that endogenously tagged SOAT1 was broadly distributed throughout the ER network, as previously reported (Lak et al, 2021). Interestingly, the enzyme became concentrated in the juxtanuclear region upon TNFα stimulation (Fig. 6E,F). In parallel, endogenous DGAT1 was not redistributed, as judged by immunostaining (Appendix Fig. S3), arguing for the specificity of the effect for SOAT1. Live cell imaging of SOAT1-SNAP in the nuclear area at 60 min of TNFα treatment showed SOAT1-SNAP typically in mobile clusters (presumably SOAT1 oligomers (Guan et al, 2020)) that moved dynamically around the generated LDs (Fig. 6G, videos available in source files). Together, these results suggest that TNFα induces a shift in SOAT1 distribution closer to the NE, where the enzyme induces the formation of CE-rich NE-LDs.

## TNFα induced nuclear translocation of NF-κB is modulated by cholesterol esterification

We next wanted to understand if the acute cholesterol redistribution induced by TNFα plays a role in the canonical pathway of TNFα induced inflammatory response or merely represents a by-product of the pathway activation. A key step in TNFα induced acute inflammatory signaling is the translocation of the transcription factor NF-κB from the cytoplasm to the nucleus, to initiate the transcription of pro-inflammatory cytokines, such as interleukin-1 β (IL-1β) (Fig. 6A). We, therefore, quantified the time-dependent nuclear translocation of NF-κB in A431 cells, based on immunostaining of the NF-kB subunit p65 (Fig. 7A). Despite some cell-to-cell variation, we found a robust and systematic increase in the nuclear p65 signal from 0 to 30 min of TNFα treatment by high-content imaging (Fig. 7B).

To assess if cholesterol esterification modulates NF-κB nuclear translocation, we compared TNFα induced NF-κB relocation in wildtype (WT), SOAT1-GFP overexpressing (SOAToe) as well as

SOAT1 inhibitor-treated cells. This showed that the TNFα-induced NF-κB nuclear entry was facilitated in SOAT1 overexpressing cells (Fig. 7C,D) and that SOAT1 inhibition slowed down NF-κB nuclear entry in both wildtype and SOAT1 overexpressing cells (Fig. 7E,F). We wondered whether the observed alteration in nuclear transport is specific to the TNFα signaling process. To test this, we investigated the nuclear translocation of the soluble photoconvertible fluorescent probe Dendra2 under conditions of altered SOAT1 activity. This showed that analogously to NF-κB, the nuclear entry of Dendra2 photoconverted in the cytoplasm was improved by SOAT1 overexpression and slowed down by SOAT1 inhibition (Fig. 7G). To summarize, these data demonstrate that cholesterol esterification modulates both the active entry of a physiological transcription factor and the passive diffusion of a model fluorescent protein to the nucleus.

## Acute SOAT1 inhibition reduces TNFα induced NE-LDs, NF-κB nuclear translocation, and transcriptional activation in macrophages

Finally, we investigated if the TNFα induced effects on cholesterol esterification and SOAT1 sensitivity of TNFα signaling can also be observed in primary innate immune cells, human monocyte-derived macrophages from blood donors. When grown in a complete macrophage culture medium, these cells harbor variable amounts of LDs that are typically TAG rich, as revealed by SRS imaging (Fig. 8A). Upon treating macrophages with TNFα for 30 and 60 min, we observed an increase in NE-LD numbers and a prominent increase of CE-storage in LDs by SRS imaging (Fig. 8A–C). In some cells, we found NE membrane tunnels traversing the nucleoplasm and harboring LDs at 60 min of TNFα treatment (Fig. EV4D). We also noticed an increased CE signal outside droplets upon TNFα treatment, suggesting increased CE levels in endomembranes, presumably in the ER (Fig. 8D,E).

The analysis of TNFα-induced NF-κB nuclear entry in macrophages showed that the NF-κB translocation was heterogeneous between individual cells but followed closely similar kinetics as in A431 cells, when analyzed in a large number of cells by high-content imaging. Remarkably, acute SOAT1 inhibition during the TNFα stimulus impaired not only the TNFα-induced formation of CE-rich LDs (Fig. EV4E–G), but also NF-κB nuclear translocation (Fig. 8F,G) and downstream IL-1β mRNA elevation (Fig. 8H). These data argue that cholesterol esterification modulates NF-κB nuclear translocation and the initiation of inflammatory signaling in macrophages.

To study if TNFα induces morphological changes in the NE, we performed focused ion beam SEM (FIB-SEM) imaging of control

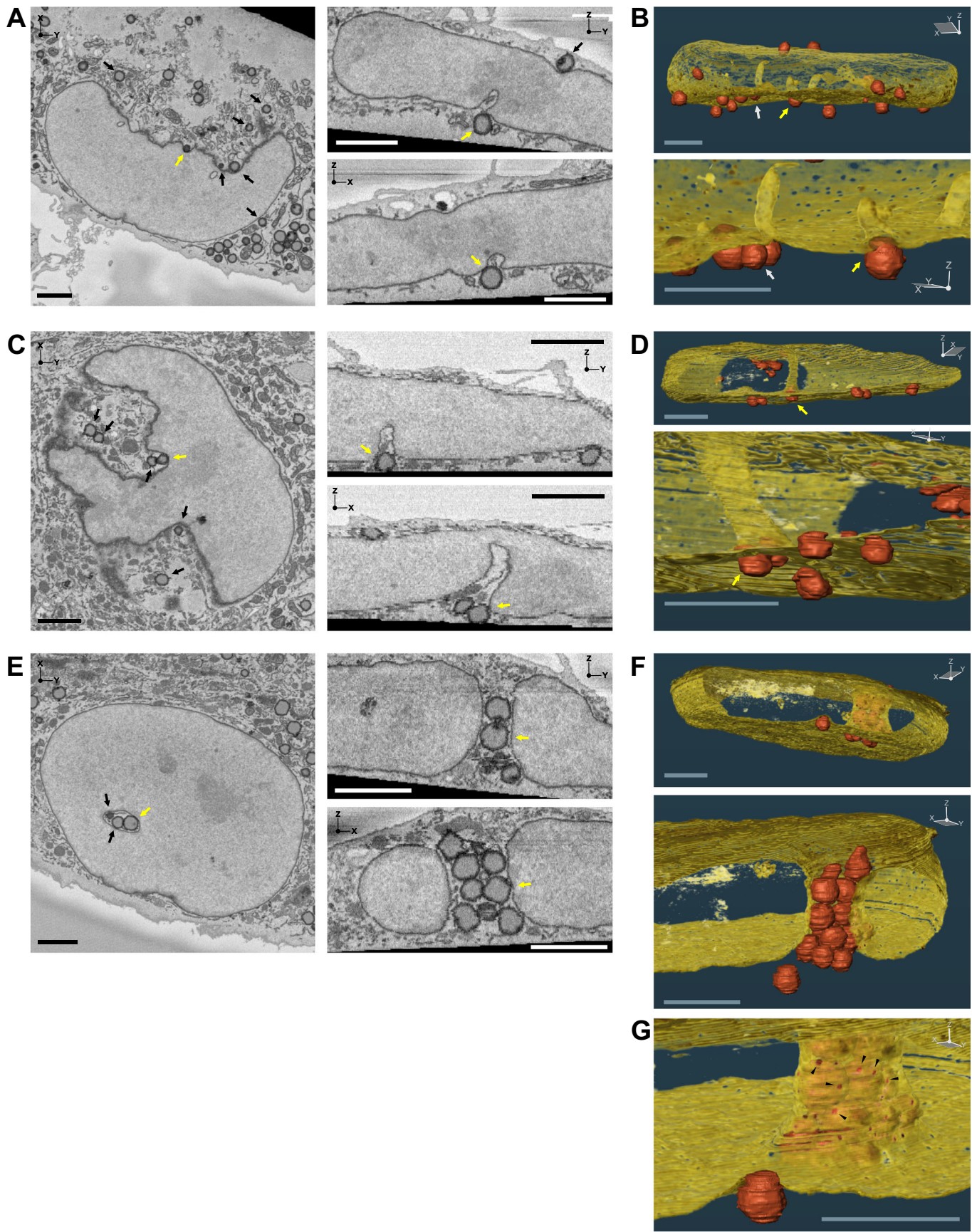

◀ **Figure 4. Individual and clustered LDs associate with NE invaginations of various extent.**

(A) Individual characteristic cross-sections in XY, ZY, and ZX planes of SBF-SEM dataset of a human macrophage cell loaded with chol and OA for 2 h, showing the formation of finger-like membrane invaginations of the NE. NE-LDs are shown with arrows; the specific NE-LD at the mouth of the invagination is highlighted with a yellow arrow. (B) The corresponding to (A) volume rendering of NE (yellow) and models of the associated NE-LDs (vermilion) highlight the 3D organization of the nucleus. (C) Characteristic cross-sections of another human macrophage cell where the NE forms a cross-nucleus tunnel from the bottom to the top of the nucleus. The cross-section layout and coloring of the arrows are the same as in (A). (D) 3D visualization of the cross-nuclear tunnel with the models of the associated NE-LDs. The same layout and coloring as in (B). (E) Third example from a human macrophage cell showing the characteristic cross-section of nucleus highlighting a wide cross-nucleus tunnel filled with LDs. The yellow arrow points to the same LD seen in all cross-sections. (F) 3D visualization of the nucleus rendering with the same colors as (B). (G) Magnified view of (F) with nuclear pores in front of LDs (arrowheads). Scale bars 2 μm. Additional views in Fig. EV3. Source data are available online for this figure.

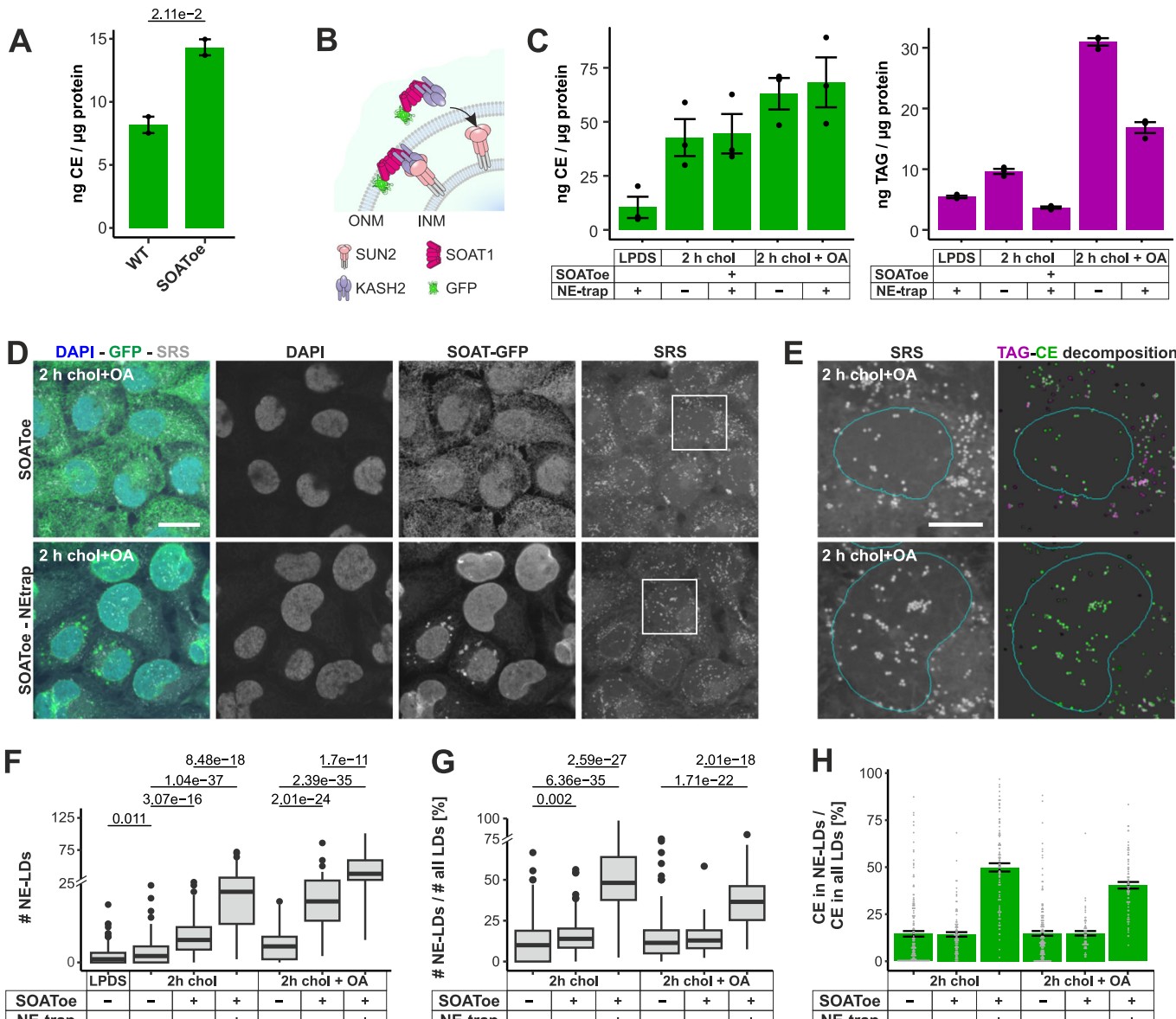

**Figure 5. SOAT1 localization contributes to the location of CE storage.**

(A) TLC analysis of wildtype and SOAT1-GFP overexpressing (SOAToe) A431 cells grown in complete medium (mean ± SEM). (B) Schematic illustration of the SUN-KASH system used to trap SOAT1-GFP to the NE. (C) TLC analysis of SOAToe and SOAToe-NE-trap cells loaded with chol or chol+OA for 2 h (mean ± SEM). (D) Representative fluorescence and SRS images of SOAToe and SOAToe-NE-trap cells loaded for 2 h with chol+OA. (E) Insets of SRS images in (D) and neutral lipid decomposition. (F, G) Number of NE-LDs and fraction of NE-LDs of all cellular LDs in WT, SOAToe and SOAToe-NE-trap cells lipid loaded as indicated. p values from Mann–Whitney U-tests, boxplots according to ggplot (see Methods). (H) Fraction of CE SRS intensity in NE-LDs over CE SRS intensity in all LDs (mean per cell ± SEM). Cell numbers: 136 for LPDS condition in WT cells (F only), 177, 99, 102 for chol load, and 164, 66, 79 for chol + OA load for WT, SOAToe, and SOAToe-NE-trap cells in (F–H). Scale bars: 25 μm (D), 10 μm (E). Source data are available online for this figure.

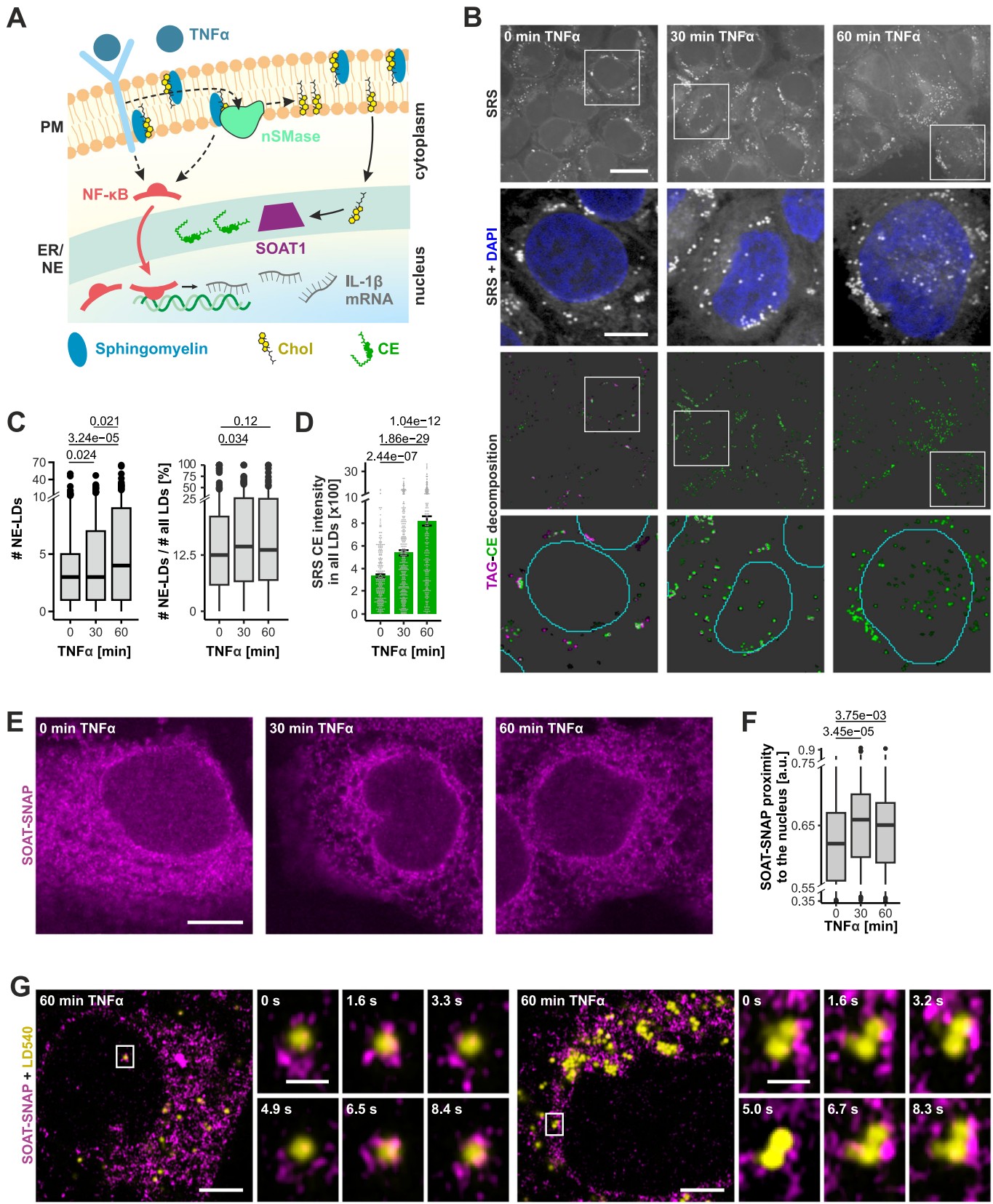

◄

**Figure 6.  TNFα induces cholesterol esterification and subcellular redistribution of SOAT1.**

(A) Schematic illustration of plasma membrane (PM) cholesterol liberation, NF-κB nuclear translocation, and upregulation of IL-1β transcript levels upon TNFα treatment. (B) Exemplary SRS images and signal decompositions of A431 cells at 0, 30, and 60 min TNFα treatment, with nuclear areas indicated. (C) Number of NE-LDs and fraction of NE-LDs over all LDs upon TNFα treatment. $p$ values from Mann–Whitney $U$-tests, boxplots according to ggplot (see Methods). (D) Decomposed CE SRS intensities from LDs upon TNFα treatment (mean per cell ± SEM). $p$ values from $t$-tests. (E) Exemplary confocal images showing the subcellular distribution of endogenous SOAT1-SNAP at 0, 30, and 60 min TNFα treatment. (F) Quantification of nuclear proximity of the SOAT1 signal. $p$ values from $t$-tests and boxplots according to ggplot (see Methods). (G) Live imaging of SOAT1-SNAP cells at 60 min TNFα treatment. The insets show snapshots of two exemplary videos from the areas indicated. Cell numbers: 277, 370, 291 in (C, D); 166, 162, 161 pooled from two biological replicates in (F) for 0, 30, and 60 min TNFα treatment. Scale bars: 25 μm (B—overview), 10 μm (B—inset, E), 5 μm (G—overview), 1 μm (G—inset). Source data are available online for this figure.

and TNFα treated macrophages, using correlative LM to identify a TNFα treated cell with NE-LDs (Fig. EV5A,B). FIB-SEM datasets collected at $6 \times 6 \times 12$-nm voxel resolution did not reveal obvious differences between the cells in the nuclear shape, NE membrane morphology, nuclear pore number or pore density that could explain how TNFα would facilitate nuclear entry (total of 1919 nuclear pores with 6.45 pores/μm$^2$ in control and 1724 pores with 4.97 pores/μm$^2$ in TNFα treated cell). Intriguingly, a difference in the lateral organization of the pores was noted, with the control cell showing a highly uniform and the TNFα treated cell a more clustered distribution of pores (Fig. EV5C,D). To scrutinize if similar changes can be observed in other systems, we challenged A431 cells expressing endogenously tagged Nup93-GFP (Li et al, 2024) with TNFα and analyzed nuclear pore distribution similarly as in macrophages. This showed a time-dependent increase in the number of nuclear pore clusters during 0–30 min of TNFα treatment (Fig. EV5E,F). Remarkably, this effect was abrogated if SOAT1 inhibitor was added together with TNFα (Fig. EV5E,F). Together, these results suggest that the TNFα induced lateral redistribution of nuclear pores in the membrane is linked to the TNFα induced cholesterol esterification.

## Discussion

In this study, we established an SRS imaging-based method for assessing CE and TAG distribution and levels in human cells. The principle of using SRS microscopy to distinguish lipids based on their vibrational properties is well established, and various strategies have been taken for signal decomposition and quantification (Wang et al, 2013; Fu et al, 2014; Jaumot et al, 2015; Hislop et al, 2022; Xu et al, 2024). Recent algorithmic developments have increased chemical specificity in lipid quantification but—due to the normalization of spectra during preprocessing—are limited to relative concentrations (Zhang et al, 2024). Here, we took an approach of directly decomposing the SRS signal from the wavenumber region characteristic to lipids into a non-negative linear mixture of predefined reference spectra recorded from cells of known neutral lipid composition. This enabled us to employ the specific SRS signal intensity as a proxy of lipid amount. Importantly, to our knowledge, no direct comparisons to biochemically measured lipid levels in the biological materials imaged have been made in previous SRS-based reports.

Using our method, we found that the CE and TAG distribution between individual LDs varied within a cell and that there were cell-type-specific characteristics in this distribution. Most strikingly, we observed that the LDs associated with the NE—that we refer to as

NE-LDs—were relatively CE enriched in all the cell types analyzed. The dynamic localization of SOAT1 in the ER with a fraction enriched in the NE, as also previously reported by us (Lak et al, 2021), is in line with this notion. Indeed, we observed that higher SOAT1 levels and activity increased NE-LDs, and this was further aggravated by the artificial trapping of active SOAT1 to the NE.

Interestingly, early biochemical analysis of LDs isolated from rat liver reported that cytoplasmic LDs were TAG enriched and LDs isolated from nuclei were CE enriched (Layerenza et al, 2013). However, the origin and roles of this potential nuclear CE pool have remained unknown. Nuclear LDs, their constituents, and topologies have since received considerable attention, but most studies have focused on phospholipid and DAG-TAG homeostasis and their regulation (Fujimoto, 2024). Moreover, nuclear LDs are considered to bud from the inner nuclear membrane, potentially from type I nuclear reticulum invaginations, and project towards the nucleoplasm.

By EM, we found the CE-rich NE-associated LDs to localize on the cytoplasmic side of the NE, often associated with NE invaginations. These data point to the possibility that this CE pool regulates the properties of the NE and that NE-LDs might be related to the formation of NE invaginations/tunnels. Of note, with rapid lipid diffusion in the membrane, a postulated CE enrichment in the NE might necessitate diffusion barrier(s), despite local synthesis. Interestingly, a recent cryo-electron microscopy study showed that in mammalian cells, ER-NE junctions have narrow constrictions that were suggested to act as barriers for e.g., lipid transport (Bragulat-Teixidor et al, 2024).

We used TNFα induced NF-κB nuclear translocation as a test case to measure if physiological communication between the cytoplasmic and nuclear compartments can be fine-tuned by manipulating cholesterol esterification. This was found to be the case not only in genetically SOAT1-engineered cells, but also in primary human macrophages, where SOAT1 was acutely inhibited only when TNFα induced cholesterol redistribution from the plasma membrane and NF-κB nuclear entry took place. Collectively, our findings imply that cholesterol esterification facilitates TNFα induced NF-κB translocation. Considering that diverse signaling processes activate plasma membrane neutral sphingomyelinase (Shamseddine et al, 2015), cholesterol esterification might represent a more common strategy to fine-tune nuclear entry.

Remarkably, our observation that also passive diffusion of a fluorophore from the cytoplasm to the nucleus was enhanced by cholesterol esterification and slowed down by its inhibition suggests that NE permeability via nuclear pore complexes (NPCs) is modulated by the membrane lipid status. As manipulation of cholesterol esterification changes the balance between unesterified and esterified cholesterol, local changes in both free cholesterol and CE pools in the NE may contribute to the observed effects.

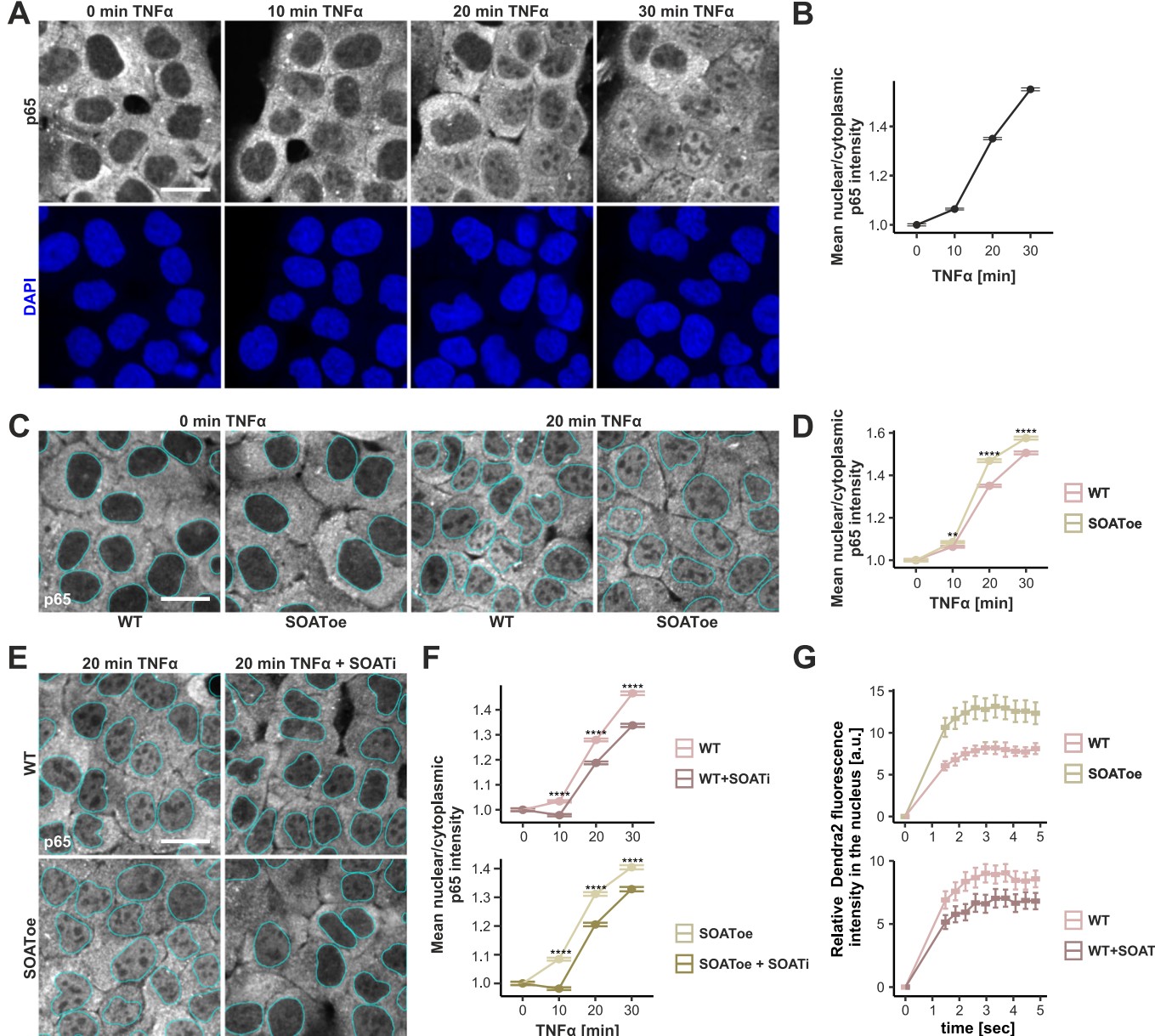

**Figure 7. SOAT1 activity affects the speed of nuclear translocation in A431 cells.**

(A) Confocal images of NF-κB nuclear translocation in A431 cells treated with TNFα for indicated times, detection with antibody against NF-κB subunit p65 and DAPI (nuclei). (B) Quantification of p65 nuclear translocation, measured as anti-p65 mean nuclear/cytoplasmic intensity relative to 0 min treatment (mean per cell ± SEM). (C) Comparison of p65 nuclear translocation in WT and SOAToe cells. Representative images from 0 and 20 min TNFα treatment. (D) Quantification of p65 nuclear translocation in WT and SOAToe cells, similarly as in (B). Exact $p$ values from $t$-tests: $9.33e - 3$, $6.93e - 48$, $8.53e - 18$ for 10, 20, 30 min TNFα treatment. (E) Effect of SOAT1 inhibitor (SOATi, 5-day pretreatment) on p65 nuclear translocation upon 20 min TNFα treatment in WT and SOAToe cells. (F) Quantification of p65 translocation in WT and SOAToe cells ± SOAT1 inhibition, similar to (B). Exact $p$ values from $t$-tests: $8.77e - 12$, $2.58e - 30$, $2.17e - 57$ for the WT (upper panel), and $4.72e - 33$, $1.27e - 33$, $3.79e - 18$ for the SOAToe (lower panel) for 10, 20, 30 min TNFα treatment, respectively. (G) Nuclear translocation of cytoplasmic, photoconverted Dendra2. Cell numbers per condition: range between 8887–9649, pooled from three biological replicates in (B); between 3165–4530, pooled from two biological replicates in (D); between 5098–7774, pooled from two biological replicates in (F); 24 and 30 for WT and SOAToe cells, and 35 and 33 for WT and WT + SOATi cells in (G). Scale bars: 25 μm (A, C). Source data are available online for this figure.

Interestingly, recent data reveal that lipid saturation can control NPC integrity and nucleocytoplasmic transport in yeast (Romanauska and Köhler, 2023). Yet, at present little is known about the interplay of NPCs and lipids at a mechanistic level (Stankunas and Köhler, 2023).

The generation of CEs in the NE, together with the finding that the ER can hold CEs to considerable levels (10–15 mol% compared to 2–3 mol% of TAGs (Dumesnil and Vanharanta et al, 2023)) argues that not only NE-LDs but also the NE may have a high CE content. This idea is supported by our SRS imaging data

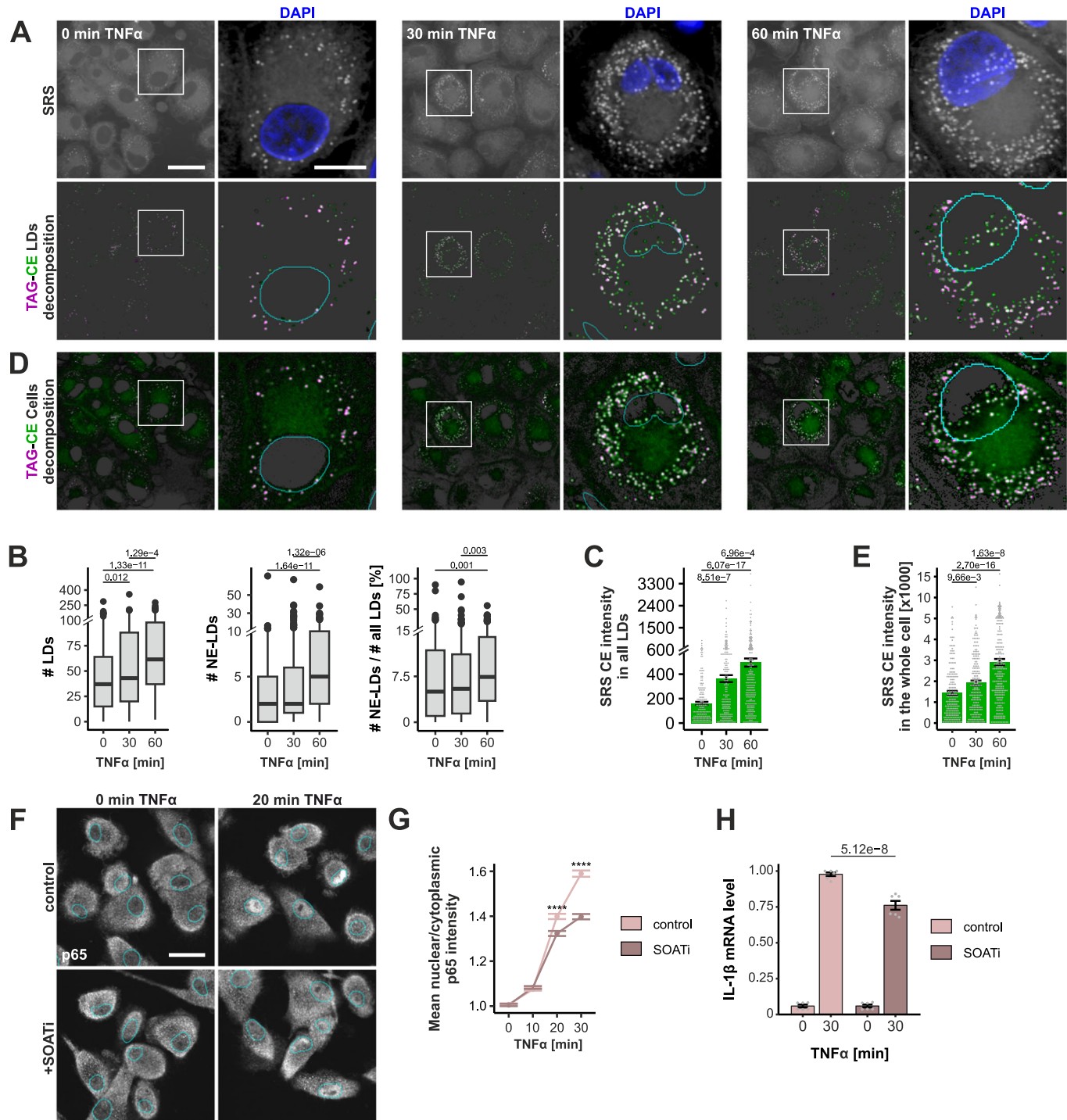

Figure 8.    Acute SOAT1 inhibition impairs nuclear translocation of p65 upon TNFα treatment in human primary macrophages.

(A) Exemplary SRS images and signal decomposition of human primary macrophages at 0, 30, and 60 min TNFα treatment, with nuclear areas indicated (DAPI staining in SRS inset, contours in decompositions). (B) Number of all LDs and NE-LDs and fraction of NE-LDs over all LDs, p values from Mann–Whitney U-tests, boxplots according to ggplot (see Methods). (C) Quantification of SRS CE intensity in all LDs, (mean per cell ± SEM). p values from t-tests. (D) Exemplary SRS images and signal decomposition from whole cells, and (E) SRS CE intensity quantification from whole cells upon TNFα treatment, (mean per cell ± SEM). p values from t-tests. (F) Representative confocal images of p65 immunostaining in primary human macrophages in control and SOAT1 inhibited conditions at 0 and 20 min of TNFα treatment. SOAT1 inhibitor was added together with TNFα. (G) Quantification of (F): p65 mean nuclear/cytoplasmic intensity upon TNFα treatment relative to 0 min TNFα in primary human macrophages (mean per cell ± SEM). Exact p values from t-tests: 1.81e − 8, 1.91e − 46 for 20, 30 min TNFα treatment. (H) IL-1β mRNA levels at 0 and 30 min ± SOATi. Pooled data from macrophages of two independent donors with triplicate measurements, p value from t-test. Cell numbers per condition: 205, 209, 252 for 0, 30, 60 min TNFα in (B, C, E); range between 2565–2752, pooled from three independent donors in (G). Scale bars: 25 µm (A—overview, F), 10 µm (A—inset). Source data are available online for this figure.

demonstrating that cholesterol loading or TNFα induced cholesterol mobilization not only resulted in CE-rich LDs but also increased a diffuse perinuclear CE SRS signal in the neighboring endomembranes. A better understanding of how CEs dynamically partition between membranes and LDs and affect their properties will be needed to elucidate how CE and CE-rich NE-LDs control the properties of the NE as a semipermeable barrier.

One can envisage that CEs might exhibit direct effects on NPC permeability e.g., via conformational plasticity of nucleoporins (Nups), as the inner ring positioned at the fusion point of the inner and outer nuclear membranes is flexible (Schuller et al, 2021). Interestingly, several Nups positioned close to the NE bilayer were identified as potentially cholesterol interacting (Hulce et al, 2013). Moreover, membrane cholesterol-CE balance might modulate the formation of NE invaginations/tunnels harboring NPCs and thereby affect cargo transfer between the cytoplasmic and nucleoplasmic space. It is also plausible that individual pores form functional assemblies in a lipid-dependent manner. Indeed, recent observations suggest that NPCs with the largest diameters formed clusters, apparently modulated by membrane tension (Morgan et al, 2024).

In conclusion, by combining SRS and CLvEM imaging, this study identifies a new subpopulation of LDs that are enriched in CE and associated with the cytoplasmic phase of the NE and with type II NE invaginations or tunnels. These NE-LDs are increased in number by exogenous cholesterol loading or by mobilization of plasma membrane cholesterol, e.g., during inflammatory signaling. Cholesterol esterification takes place in the NE and facilitates, by mechanisms to be identified, the regulated nuclear entry of a transcription factor as well as passive diffusion of cargo into the nucleus.

# Methods

### Reagents and tools table

| Reagent/resource | Reference or source | Identifier or catalog number |
|---|---|---|
| **Experimental models** | | |
| A431 cells (*H. sapiens*) | ATCC | Cat# CRL-1555, RRID:CVCL_0037 |
| Human primary macrophages | Finnish Red Cross | |
| A431 NUP93-GFP | Li et al, 2024 | PMID: 38409044 |
| A431 SOAT1-GFP oe | this study | |
| A431 SOAT1-GFP oe NE-trap | this study | |
| A431 SOAT1-SNAPf | this study | |
| **Recombinant DNA** | | |
| pDendra2-C1 | Prof. Vladislav Verkhusha, Albert Einstein College of Medicine | |
| **Antibodies** | | |
| p65 | Abcam | AB16502 |
| DGAT1 | Santa Cruz | sc-271934 |
| anti-rabbit IgG (H + L) secondary antibody, Alexa Fluor™ 647 | Thermo Fisher | A31573 |
| SOAT1 | Santa Cruz | sc-137013 |

| Reagent/resource | Reference or source | Identifier or catalog number |
|---|---|---|
| **Oligonucleotides and other sequence-based reagents** | | |
| SUN2 cDNA | https://www.ncbi.nlm.nih.gov/nuccore/ | NM_015374.3 |
| NucleoSpin RNA isolation kit | Macherey-Nagel | Cat. 740955 |
| **Chemicals, Enzymes and other reagents** | | |
| DAPI | Sigma-Aldrich | D9542 |
| DGAT1 inhibitor | Sigma-Aldrich | PZ0207 |
| DGAT2 inhibitor | Sigma-Aldrich | PZ0233 |
| SOAT1 inhibitor | Sigma-Aldrich | Sandoz 58-035 |
| Fibronectin | Roche | 11080938001 |
| LD540 | Princeton BioMolecular Research | |
| TNFα | Sigma-Aldrich | T0157 |
| TNFα | RnD Systems | 210-TA |
| methyl-β-cyclodextrin | Sigma-Aldrich | C4555 |
| cholesterol | Sigma-Aldrich | C8667 |
| NucRed Live 647 | Molecular Probes Life Technologies | R37106 |
| glutaraldehyde | Sigma-Aldrich | |
| oleic acid | Sigma-Aldrich,I | O1008 |
| SNAP-Cell 647-Sir | New England Biolabs® | S9102 |
| **Software** | | |
| Matlab R2020a | https://mathworks.com/products/matlab.html | RRID: SCR_001622 |
| R 4.4.0 | https://www.r-project.org/ | RRID: SCR_001905 |
| BIAS | Single-Cell Technologies Ltd https://single-cell-technologies.com/bias-2/ | |
| Object Analyzer | https://bitbucket.org/szkabel/lipidanalyser/downloads/ | |
| Microscopy image browser | https://mib.helsinki.fi/ | RRID: SCR_016560 |
| **Other** | | |
| SRS microscope | Tomberg et al, 2024 | https://doi.org/10.1364/OPTCON.532676 |

# Materials

Cell culture reagents and general reagents were from Gibco/Thermo Fisher, Lonza, and Sigma-Aldrich. DAPI (Sigma-Aldrich, D9542), DGAT1 inhibitor (Sigma-Aldrich, PZ0207), DGAT2 inhibitor (Sigma-Aldrich, PZ0233), SOAT1 inhibitor (Sigma-Aldrich, Sandoz 58-035), Fibronectin (Roche, 11080938001), LD540 (Princeton BioMolecular Research), TNFα (Sigma-Aldrich, T0157 and RnD Systems, 210-TA). Lipoprotein-deficient serum (LPDS) was made from fetal bovine serum (FBS) as previously described (Goldstein et al, 1983). Methyl-β-cyclodextrin (Sigma-Aldrich, C4555) and

cholesterol (Sigma-Aldrich, C8667) were mixed to prepare a cholesterol/methyl-ß-cyclodextrin complex. Methyl-ß-cyclodextrin was dissolved in $H_2O$ and incubated with cholesterol for 30 min in a sonicator at 37 °C. The resulting stock solution (50 mM cholesterol/methyl-ß-cyclodextrin complex in ratio 1:6.2) was filtered using a 0.2 μm syringe filter. BSA (Sigma-Aldrich, A3803) and oleic acid (Sigma-Aldrich, O1008) were mixed to prepare an oleic acid/BSA 8:1 complex. Oleic acid was dissolved in $H_2O$ at 37 °C before mixing it drop wise with BSA dissolved in serum-free DMEM. The resulting solution was filtered using a 0.2 μm syringe filter.

## Cell culture and lipid manipulations

A431 cells (ATCC, Cat# CRL-1555, RRID:CVCL_0037), KGN cells, and U2OS cells were cultured in Dulbecco's Modified Eagle's Medium (DMEM) containing 10% FBS, L-glutamine (2 mM), and penicillin/streptomycin (100 U/mL each) at 37 °C in 5% $CO_2$. Cell lines were regularly tested negative for mycoplasma infection by PCR. Human buffy coats were acquired from the Finnish Red Cross (permit number 44/2023), in accordance with the WMA Declaration of Helsinki and isolated and differentiated as previously described (Bøyum, 1964; Gungor et al, 2019). Where indicated, cells were delipidated in serum-free medium supplemented with 5% LPDS and DGAT1&2 inhibitors (5 μM each) for 18 h before washing twice with PBS and lipid loading. Cells were loaded with 200 μM methyl-ß-cyclodextrin complexed cholesterol and 200 μM BSA complexed oleic acid in 5% LPDS (A431, KGN, U2OS) or differentiation medium (macrophages) as indicated for each experiment. SOAT1 inhibitor was used (2 μg/ml) where indicated. TNFα (10 ng/mL) was added to cells in complete medium (A431) or differentiation medium (macrophages) as indicated.

## Generation of SOAT1 overexpressing cell lines

All overexpressing cDNA sequences were attached in Appendix Supplementary Methods. To overexpress SOAT1 with or without NE trapping in A431 cells, endogenous SOAT1 was first knocked out through CRISPR/Cas9 in A431 cells (Ran et al, 2013). A single clone was isolated and tested to be deficient in cholesterol esterification. This clone was used to generate SOAT1 over-expressing cell lines. SOAT1 with a C-terminal sfGFP tag was overexpressed in the vector pEF-IRES-P, and cells were selected with puromycin for 1 week, followed by single-cell cloning. To generate SOAT1-NE-trapped cells, SOAT1.sfGFP.KASH was over-expressed in the same vector and a single clone with a GFP level similar to the SOAT1.sfGFP cells was isolated. SUN2 cDNA (NM_015374.3) was further overexpressed in the vector pcDNA4/HisMax C in the NE-trap cell line, as the trapping was not efficient without the overexpression. The NE-trapped cells were then selected with Zeocin and single-cell cloning.

## Generation of endogenously tagged cell lines

The HDR template sequence for SOAT1-SNAPf is provided in Appendix Supplementary Methods. To generate endogenously tagged SOAT1-SNAPf A431 cell lines, CRISPR/Cas9 technology was used (Ran et al, 2013). The HDR template plasmid was transfected into A431 cells together with a vector encoding Cas9, sgRNA targeting the C-terminus of SOAT1 locus (sgRNA target site, CGTTACGTGTTTT AGAAGCT tgg) and puromycin selection marker. Clones were

isolated after transient selection with puromycin. Homozygously tagged cell lines were validated by genomic PCR and Western blot with an antibody against the endogenous SOAT1 (Mouse anti-SOAT1 (D-1), Santa Cruz sc-137013). A431 cells expressing endogenous NUP93-GFP were generated as previously described (Li et al, 2024).

## Thin-layer chromatography

For neutral lipid analysis, cells were collected in 2% NaCl, and lipids were extracted as described (Bligh and Dyer, 1959). Equal amounts of each sample were separated by thin layer chromatography (TLC) using a running solvent of hexane: diethyl ether: acetic acid (80:20:1). CE and TAG amounts were quantified from charred TLC plates, using standards for CE and TAG, in Fiji.

## SRS microscopy

The construction and performance of the in-house developed multimodal SRS microscope has been described (Tomberg et al, 2024). In brief, the instrument is composed of an inverted Olympus FV3000 confocal laser scanning microscope (Olympus, Japan), coupled with a dual-output femtosecond laser source (InSight X3+, Spectra-Physics, MKS Instruments, USA) providing a wavelength tunable pump beam and a fixed Stokes beam at 1045 nm for SRS microscopy. The pump and Stokes beams are passed through a spectral focusing, timing, and recombination unit (SF-TRU, Newport, USA) before coupling to the FV3000 microscope. The light was focused on samples with UPLSAPO 60xW1600 objective (Olympus, Japan), collected in transmission mode with an oil immersion condenser (Leica Microsystems, Germany), and detected with an SRS Detection set (APE Angewandte Physik & Elektronik Berlin, Germany). Hyperspectral SRS images were acquired with varying total laser power and zoom factor 1x or 2x for optimized signal level. The pump/Stokes ratio was held approximately the same at about 1:3 ratio while the individual power levels varied for three different pump wavelengths of 890 nm, 805 nm, and 795 nm for Raman shift ranges of 1625–1775, 2800–2950, and 2950–3100 $cm^{-1}$, as listed in Appendix Table S1. The step size in these spectrally focused hyperspectral SRS images was 3 $cm^{-1}$, resulting in SRS images with 153 wavenumbers and with bit-depth 12. Parallel to the SRS images, fluorescent images of DAPI staining and SOAT1-GFP were acquired using 405 and 488 nm excitation lasers, respectively.

## Segmentation of nuclei and LDs in fluorescence and SRS images

Nuclei were segmented based on DAPI images with Biological Image Analysis Software (BIAS) Lite (Single-Cell Technologies Ltd. v1.1.0) with its generic-nucleus segmentation deep learning model (Mund et al, 2022; Hollandi et al, 2020). The identified nuclei were manually curated in case of segmentation errors. For segmentation and visualization, the hyperspectral SRS images were collapsed by mean projection after subtracting experiment-specific cover glass background individually from each pixel. The subtraction was capped at 0 to remain in the space of non-negative numbers. The cover glass background was defined by identifying the 2000 pixels with the lowest mean SRS-signal intensity from each image and averaging their spectra to form a single background spectrum. These background subtracted and mean-

collapsed images are used throughout the study for visualizing SRS signals in exemplary images. Cells were segmented with a modified watershed algorithm (Propagation in the CellProfiler module Identify-Secondary) around the identified nuclei, using the mean-collapsed SRS images. Lipid droplet identification was performed on the collapsed SRS images with the DetectAllSpots module described earlier (Salo et al, 2019). Briefly, the module is segmenting droplets based on a multi-level à trous wavelet transform to detect droplets combined with thresholding to identify droplet instances. The segmentation pipeline was performed with custom software written in Matlab, including parts of CellProfiler (Carpenter et al, 2006), and is freely available at: https://bitbucket.org/szkabel/lipidanalyser/downloads/. NE-LDs were defined as droplets whose segmentation mask was overlapping with the nuclear mask, including partial overlap.

## Deconvolving mixed SRS signal to lipid components

The hyperspectral image stacks consisted of 153 dimensions (wavenumbers). To maintain signal characteristics minimal pre-processing was performed. Each pixel was decomposed to the non-negative linear combination of four reference spectra (components) as follows. TAG and CE reference spectra were collected from manually selected LDs of only OA or cholesterol-loaded cells (5–12 droplets per load, Fig. EV1A,B). Selected droplets' spectra were individually checked for characteristics of the corresponding lipid class (i.e., peaks highlighted in Fig. 1C). Cellular background reference spectrum was collected from manually defined cytoplasmic regions of lipid-starved cells without load (4–7 regions, Fig. EV1A,B). Additionally, to adjust for the signal originating from water and cover glass, we defined the water background spectrum from areas of images without any cells (2–4 regions, Fig. EV1A,B). The reference spectrum for each of these components was defined as the mean of the collected pixels. To remove the cellular membrane signal, we subtracted the cellular background component from the reference spectra of TAG and CE (Fig. EV1B). The subtraction was capped at 0 to keep the signal non-negative. Notably, we did not subtract the cellular background from the spectra of the pixels under investigation, because including the cellular background component in the linear model inherently includes a pixel-wise scaled background subtraction. The obtained reference spectra were column-wise placed into a $153 \times 4$ matrix $S$. All pixels to be decomposed were row-wise collected into matrix $D$ with dimensions $N \times 153$ ($N$ is the number of pixels), and by using Matlab's lsqnonneg function, we identified the component weight matrix $W \geq 0$ with dimensions $N \times 4$ such that $||WS^T - D||_F$ is minimal. We used the identified weights directly as estimates for lipid amount at the given location and referred to them as SRS lipid intensity (Fig. EV1C).

## Visualizing lipid intensities by SRS imaging

SRS lipid intensities were back-projected to their pixel location to produce false-colored images with magenta referring to TAG and green referring to CE. On the false-colored images, pixels falling outside of the segmented areas were colored gray. For visualization purposes only, the SRS lipid intensities were adjusted to the range 0–0.2/0.3/0.4/0.5 (where 1 refers to the maximum lipid intensity of the visualized components), exact adjustment limits are available in the source data files. This approach reduced the effects of imaging artefacts that caused aberrantly high lipid intensities in very few pixels and enabled seeing lipid distribution in pixels with lower SRS intensities.

## Quantification of lipid intensities based on SRS imaging

Prior to quantification, all the component weights (including SRS lipid intensities) were linearly adjusted to the most common imaging settings with respect to the zoom factor and laser power. The laser power parameters were kept identical across the images used for reference generation. In addition to careful power settings, pixels saturated due to the lock-in detector signal saturation were filtered out. We termed a pixel saturated if, in its spectrum, the standard deviation over a $12 \text{ cm}^{-1}$ wavenumber range was below 50, and at the same wavenumber, the SRS value was above 3500 (85% of the possible maximum). On the false-colored SRS lipid intensity images, saturated pixels were colored gray.

Following the adjustments, the CE content (fraction of CE) of every LD was defined by summing up the CE and TAG SRS lipid intensities from the pixels belonging to the LD (denoted by $CE_{LD}$ and $TAG_{LD}$ and forming the ratio: $CE_{LD}/(CE_{LD} + TAG_{LD})$. Only LDs with less than 20% saturated pixels were included in the quantifications. The cell-wise mean SRS lipid amounts were formed by summing up SRS lipid intensities from all LD pixels (for LD quantifications)/all cellular pixels (for cellular quantifications) and dividing them by the number of cells.

As the scales of biochemical (TLC) lipid analysis and SRS lipid intensities are not directly comparable, the mean value of CE amount in double (cholesterol and OA) loaded cells was used as a common normalization ground to form the artificial scale of lipid amount when plotting SRS and TLC results together. Consequently, in the joint TLC-SRS plots the CE levels of the double load are matched artificially, but all other matches present real similarities between the quantification techniques.

## Cell stainings for fluorescence and SRS microscopy

For SRS imaging, cells were seeded on fibronectin (10 μg/mL)-coated coverslips; for high-content imaging of p65 immunofluorescence, cells were seeded on 384-well imaging plates (Corning, 4518) and for intracellular distribution analysis of DGAT1, SOAT1-SNAP and NUP93-GFP, cells were seeded on fibronectin (10 μg/mL) -coated 8-well imaging chambers (Thermo Fisher Scientific, 155409). Cells were treated as indicated for each experiment in figure legends, washed with PBS, and fixed with 4% PFA (or 2% PFA for imaging of SOAT1-SNAP) in 250 mM Hepes, pH 7.4, 100 μM CaCl₂, and 100 μM MgCl₂ for 20 min and washed with PBS. For SRS imaging, coverslips were stained with DAPI (5 ug/mL) in PBS for 10 min, mounted on microscopy slides in $H_2O$, and sealed with epoxy glue. For immunofluorescent imaging, cells were quenched in 50 mM NH₄Cl for 10 min, washed with PBS, permeabilized in 0.1% Triton X-100 (for p65 staining) or 0.1% saponin (for DGAT1 staining) in PBS for 10 min, washed with PBS and blocked in 1% BSA in PBS + 0.1% Tween-20 for 1 h. Subsequently, cells were incubated in the blocking solution containing the p65 primary antibody (Abcam, AB16502, 1:500) or DGAT1 primary antibody (Santa Cruz, sc-271934, 1:100) and washed three times for 5 min with PBS + 0.1% Tween-20 before incubating in the blocking solution with the Alexa Fluor 647 conjugated secondary antibody (Thermo Fisher, A31573, 1:200) for

1 h, incubation for 10 min in DAPI (5 μg/mL) in PBS, and washing three times 5 min in PBS. p65 immunofluorescence was imaged with a Perkin-Elmer Opera Phenix automatic spinning-disk confocal microscope using 63x water objective with NA 1.15. DGAT1 and SOAT1-SNAP were imaged with Airyscan Zeiss LSM 880 using Plan-Apochromat 63x/1.4 Oil DIC M27 objective. NUP93-GFP was imaged with Leica TCS SP8 CARS and multi-stack images of NUP93-GFP were maximum intensity projected.

## Live cell imaging of SOAT1-SNAP

For monitoring the interactions between SOAT1 and LDs in live cells, SOAT1-SNAP expressing cells were labeled with 120 nM SNAP-Cell® 647-SiR substrate (NEB, S9102) for 5 min, washed three times with culture medium and incubated in culture medium for 1 h at 37 °C in 5% $CO_2$. Subsequently, cells were stained with LD540 (200 ng/mL) and treated with TNFα as indicated in the Figure legend. SOAT1-SNAP and LD540 fluorescence were imaged with Airyscan Zeiss LSM 880 using Plan-Apochromat 63x/1.4 Oil DIC M27 objective.

## Image analysis of SOAT1 and DGAT1 subcellular distribution

Nuclei and cytoplasms were identified as in SRS images with SOAT1-SNAP or DGAT1 fluorescence channels forming the basis for cell detection. Nuclear proximity of fluorescence was defined as the intensity weighted average of normalized distances, formally: $\sum_k \frac{i_k}{I} * \frac{d_{PM,k}}{d_{N,k}+d_{PM,k}}$, where $d_N$ and $d_{PM}$ denote the shortest distance to the nucleus (identified by DAPI staining) or plasma membrane (contour of the cell mask), $i_k$ is the intensity of pixel k, $I$ is the total intensity in the cell. This value is 1 for the signal in the nucleus and 0 for the signal at the edge of the cell mask (Appendix Fig. S4).

## High-throughput image analysis of p65 nuclear translocation

Multi-stack images of A431 cells and human macrophages were collapsed using maximum intensity projection and single-slice selection, respectively. In the case of macrophages, we identified the slice with the highest intensity on the DAPI channel and selected the corresponding image slice from the other channels. Nuclei and cells were segmented similarly as in the case of SRS images, with the exception that cells were identified based on the p65 channel. For human macrophages, cells smaller than a diameter of 20 μm were excluded to filter non-differentiated cells. p65 intensities were background subtracted with the well-specific outside cell average intensity. p65 translocation was measured as the ratio between the nuclear and cytoplasmic (cell—nuclear) background subtracted by mean intensity.

## Dendra2 imaging and quantification

A431 cells were transfected with pDendra2-C1 plasmid (a gift from Vladislav Verkhusha, Albert Einstein College of Medicine) 24 h before reseeding on a four-well chamber (Thermo Fisher, 155383) coated with fibronectin. The following day, nuclei were stained by NucRed Live 647 (Molecular Probes Life Technologies, R37106) for 1.5 h before Dendra2 photoconversion with Leica TCS SP8 CARS (Leica Microsystems GmbH, Germany), equipped with HC PL APO CS2 40x/1.10 water objective, at 37 °C, 5% $CO_2$. The double fluorescence (green and red) of Dendra2 was excited by 488 nm laser (5% power) and by 561 nm laser (50% power), and detected at 525/25 nm and at 590/20 nm, respectively. NucRed was excited with a 633 nm laser (45.2% power) and detected at 728.5/17.5 nm. Dendra2 was photoconverted in a cytosolic 4 μm × 4 μm region by a 405 nm laser (1.113 s, 50% power). For quantification of cytosolic Dendra2 translocation to the nucleus, Dendra2 red fluorescence measured in a 3 μm × 3 μm nuclear region was normalized by Dendra2 green fluorescence measured pre-photoconversion at the photoconverted region. Quantifications were performed using Fiji/ImageJ (version 1.54 f).

## Quantitative reverse transcription-PCR

RNA was isolated using the NucleoSpin RNA isolation kit (Macherey-Nagel, Cat. 740955) and transcribed using the SuperScript VILO cDNA synthesis kit (Invitrogen). Quantitative reverse transcription-PCR was performed with Light Cycler 480 SYBR Green I Master Mix (Roche) and a Light Cycler 480 II (Roche). Reaction steps: 95 °C for 15 min, 40 cycles of 95 °C 15 s, 60 °C 30 s, and 72 °C 10 s. Quantities of mRNAs were normalized to 18S. Primers for 18S: GTCTTGTAGTTGCCGTCGTCCTTG (forward) and GTCTTGTAGTTGCCGTCGTCCTTG (reverse), and for IL-1β: ATGATGGCTTATTACAGTGGCAA (forward) and GTCGGAGATTCGTAGCTGGA (reverse).

## Volume electron microscopy

Cells grown on glass-bottom dishes with etched grid markings (MatTek, Ashland, MA) were fixed with 2% glutaraldehyde (Sigma-Aldrich) in 0.1 M sodium cacodylate, 2 mM $CaCl_2$ buffer (pH 7.4), for 30 min at room temperature and stained with LD540 (0.1 μg/mL) in 0.1 M sodium cacodylate buffer for 20 min. LDs were imaged in 0.1 M sodium cacodylate buffer using a Zeiss LSM 880 confocal microscope with 63x Plan-Apochromat oil objective, NA 1.4. After light microscopy imaging, the fixed cells were flat embedded for SBF-SEM or FIB-SEM using a protocol modified from (Deerinck et al, 2010) as described previously (Puhka et al, 2012). The cells of interest were identified using the replicated coordination lines and trimmed from the plastic blocks for vEM imaging. SBF-SEM datasets were acquired with a FEG-SEM Quanta 250 (FEI, Hillsboro, OR) equipped with a microtome (Gatan 3View; Gatan Inc., Pleasanton, CA) using 2.5-kV beam voltage, spot size 3, pressure 0.23–0.30 Torr, dwell time 5–10 μs producing datasets with 30 nm slicing and 10 × 10 or 15 × 15 nm pixels. FIB-SEM datasets were acquired with Crossbeam 550 FIB-SEM (Zeiss, Gemany) using 1.6-kV beam voltage, 1 nA current, and 4 μs dwell time, producing datasets with 12 nm slicing and 6 × 6 nm pixels. The basic processing of images: adjustment of brightness and contrast, normalization of intensities, alignment, and the following segmentation was done using Microscopy Image Browser (MIB) (Belevich et al, 2016; Belevich and Jokitalo, 2021) The generated models were exported and further rendered in Amira (Thermo Fisher Scientific, Waltham, MA).

## Segmentation of LDs, nuclei, and nuclear pores for vEM datasets

All segmentations were conducted in MIB(Belevich et al, 2016). To segment the nucleus, we employed the semi-automatic graph-cut-based

segmentation tool, where the entire nucleus was segmented using a few manually added landmark points. Morphological operations such as 3D erosion followed by 3D dilation were used to generate a mask for the visualization of the nuclear envelope (NE) using volume rendering. Models of LDs in the vicinity of the NE were manually generated using the interactive segment-anything tool (Kirillov et al, 2023) implemented in MIB. For the detection of nuclear pores, we trained a convolutional neural network (CNN) in DeepMIB (Belevich and Jokitalo, 2021) using a 3D U-Net architecture (Çiçek et al, 2016). To generate ground-truth data for CNN training, the FIB-SEM datasets were downsampled to isotropic $12 \times 12 \times 12$-nm voxels, where on a fraction of NE (~1/6), locations of nuclear pores were manually annotated, and each pore was estimated as a sphere (diameter of 7 pixels). The CNN was trained on patches of $96 \times 96 \times 96$ pixels using a random cocktail of 16 various augmentations overnight. The application of the trained network to the FIB-SEM datasets was done using a 15% overlap between patches. The resulting models were checked and corrected to include only models of nuclear pores. Subsequently, 3D coordinates of each nuclear pore were identified using the Get Statistics tool in MIB and extracted for clustering analysis. A neighborhood graph was built based on the 3D localization of the nuclear pores with pores as nodes and edges between those pores that were closer than a predefined threshold, adjusted with the volume of the nucleus. Pores that fell into the same connected component in the neighborhood graph were considered to form a cluster.

### Segmentation of nuclear pores in NUP93-GFP A431 cells

Nuclei were identified with BIAS's deep learning nuclei segmentation model based on the Gaussian-filtered NUP93-GFP channel. For the detection of the nuclear pores, an Ilastik (Berg et al, 2019) pixel classification model was trained on the NUP93-GFP channel, followed by using the DetectAllSpots module on the classification probability image to segment nuclear pores. Cluster analysis was performed for EM with a fixed threshold and using 2D coordinates.

### Statistics and software

Figures show exact *p* values from statistical tests as indicated in Figure legends. Bar charts show mean ± standard error, and individual data points. Exact cell numbers and replicate information are provided in the source data files. Boxplots follow the specifications of ggplot: the line is the median, lower and upper hinges are the first and third quartiles, and whiskers extend to 1.5 times the interquartile range (Wickham, 2016). Image and signal analysis were performed with Matlab R2020a, plot generation, and statistical analyses with R version 4.4.0 on a standard PC with 16GB of RAM. We utilized ggbreak in plotting graphs (Xu et al, 2021).

## Data availability

Codes used to perform the SRS signal processing and decomposition in this study are freely available at: https://bitbucket.org/szkabel/srs/downloads/. The image analysis software used for LD and cell segmentation is available at: https://bitbucket.org/szkabel/lipidanalyser/downloads/. The microscopy image browser is available at: https://mib.helsinki.fi/. Materials generated in the study are available on request. Hyperspectral SRS and fluorescent images, as well as spectral decompositions shown in the figure panels, were

deposited to BioImage Archive with accession number: S-BIAD1584: https://doi.org/10.6019/S-BIAD1584.

The source data of this paper are collected in the following database record: biostudies:S-SCDT-10_1038-S44318-025-00423-2.

## Peer review information

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

## Acknowledgements

The authors thank Anna Uro (Ikonen lab) and Mervi Lindman (EMBI) for excellent technical assistance and sample preparation. We thank the Biomedicum Imaging Unit, supported by the Helsinki Institute of Life Science (HiLIFE) and Biocenter Finland, for support in light microscopy. EJ, IB, and EI acknowledge support from Euro-BioImaging Finnish Advanced Microscopy Node, Biocenter Finland, and Helsinki Institute of Life Science. EI acknowledges support from the Academy of Finland (grant 324929), Fondation Leducq (grant 19CVD04), Sigrid Juselius Foundation, and Jane and Aatos Erkko Foundation. TT and CS acknowledge support from the Academy of Finland (grants 327732 and 331837). AS acknowledges the University of Helsinki, Doctoral Program of Integrative Life Science for funding the project. LV acknowledges support from the Instrumentarium Science Foundation and The Finnish Medical Foundation. CV acknowledges support from OTKA-SNN 139455/ARRS.

## Author contributions

Ábel Szkalisity: Conceptualization; Data curation; Software; Formal analysis; Funding acquisition; Visualization; Methodology; Writing—original draft; Project administration; Writing—review and editing. Lauri Vanharanta: Conceptualization; Data curation; Formal analysis; Funding acquisition; Investigation; Visualization; Methodology; Writing—original draft; Project administration; Writing—review and editing. Hodaka Saito: Formal analysis; Investigation; Methodology; Writing—review and editing. Csaba Vörös: Formal analysis; Investigation; Methodology; Writing—review and editing. Shiqian Li: Investigation; Methodology; Writing—review and editing. Antti Isomäki: Formal analysis; Methodology; Writing—review and editing. Teemu Tomberg: Formal analysis; Methodology; Writing—review and editing. Clare Strachan: Resources; Funding acquisition; Project administration; Writing—review and editing. Ilya Belevich: Formal analysis; Investigation; Visualization; Methodology; Writing—review and editing. Eija Jokitalo: Resources; Supervision; Funding acquisition; Writing—original draft; Project administration; Writing—review and editing. Elina Ikonen: Conceptualization; Resources; Supervision; Funding acquisition; Validation; Investigation; Writing—original draft; Project administration; Writing—review and editing.

Source data underlying figure panels in this paper may have individual authorship assigned. Where available, figure panel/source data authorship is listed in the following database record: biostudies:S-SCDT-10_1038-S44318-025-00423-2.

## Disclosure and competing interests statement

The authors declare no competing interests.

# Expanded View Figures

**Figure EV1.   SRS reference spectra and whole-cell signal decomposition in A431 cells.**

(A) Exemplary SRS images at 2860 cm$^{-1}$ Raman shift with exemplary ROIs for defining reference spectra for cellular background, TAG, CE, and water background. (B) Reference spectra for the components defined in (A): two large areas for cellular background (whose spectra closely overlap), three individual LDs each for TAG and CE, and two large areas for water background (with also closely overlapping spectra). The final TAG and CE reference spectra (bottom row) are generated by subtracting the cellular background spectrum from the TAG and CE spectra measured from LDs (top row). (C) SRS lipid intensities (weight coefficients) for the ROIs indicated in (A), mean and standard error of pixel-based weights in each of the ROIs. (D, E) Exemplary false color visualization of the SRS signal decomposition of a 2 h chol loaded A431 cell. TAG and CE visualization for both LDs only (LDs) and whole cell (Cell). (F) Comparison of whole-cell lipid amounts measured by SRS and TLC at 2 h chol load (mean per cell ± SEM). Estimated SRS and TLC values are normalized similarly as in Fig. 1H. Number of pixels in (C) range from 4 to 18747 (exact numbers in source data). 290 cells in SRS and two technical replicates for TLC in (F). Scale bars: 5 μm (A—cellular background, TAG, CE), 20 μm (A—water background), 10 μm (D). Source data are available online for this figure.

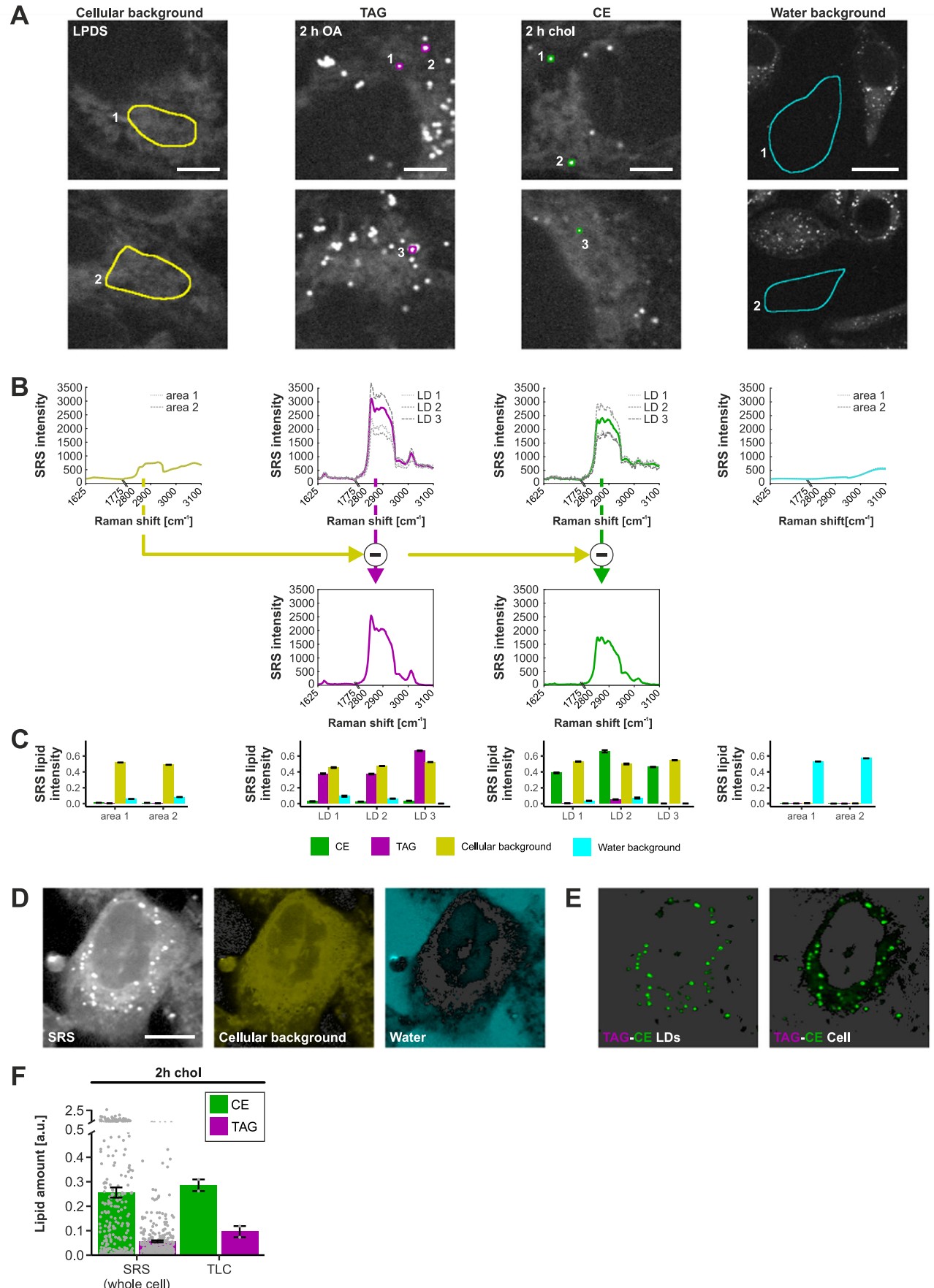

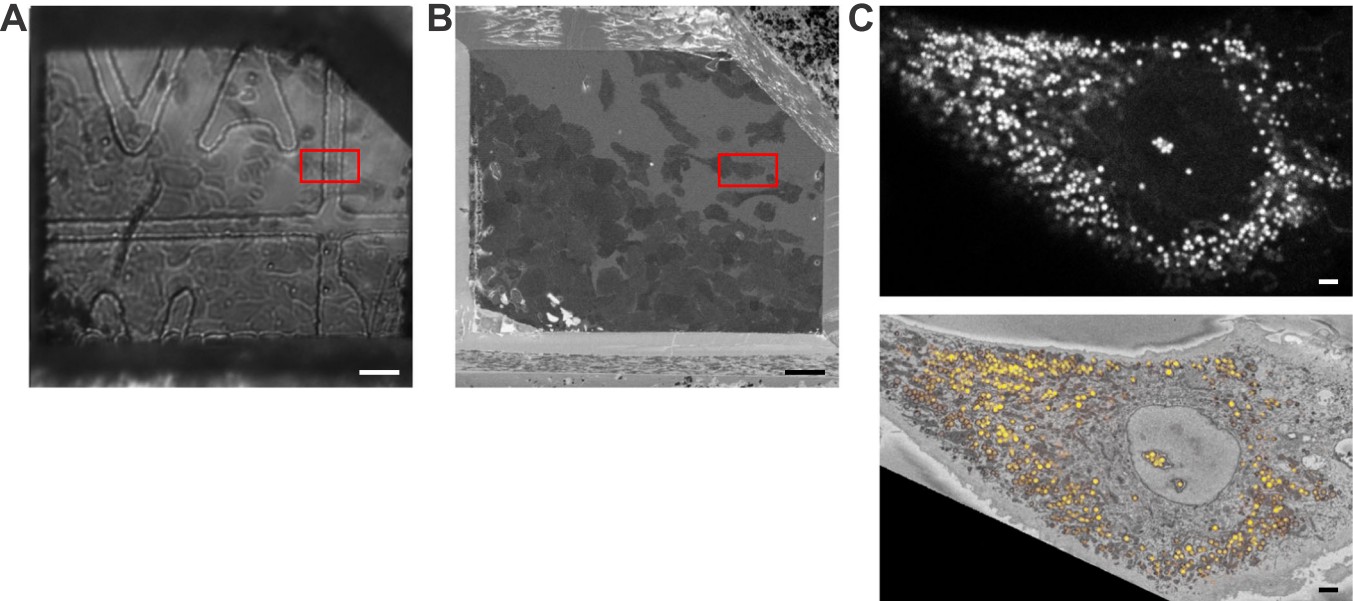

**Figure EV2. CLvEM workflow of human primary macrophages.**

(A) LM image of a block face with embedded human primary macrophages loaded with chol and OA for 2 h and prepared for CLvEM. Lines and labels on the top of the block are used to find the cell of interest for the correlative workflow. The red rectangle indicates the location of the cell of interest. (B) SEM image of the block face taken at 15 kV showing the location of cells. (C) A confocal LM image showing LDs stained with LD540 and an overlay image showing the correlation between LM and SBF-SEM datasets. The SBF-SEM signal is visualized in grayscale and LM signal in yellow. Scale bars: 80 µm (A, B), 2 µm (C).

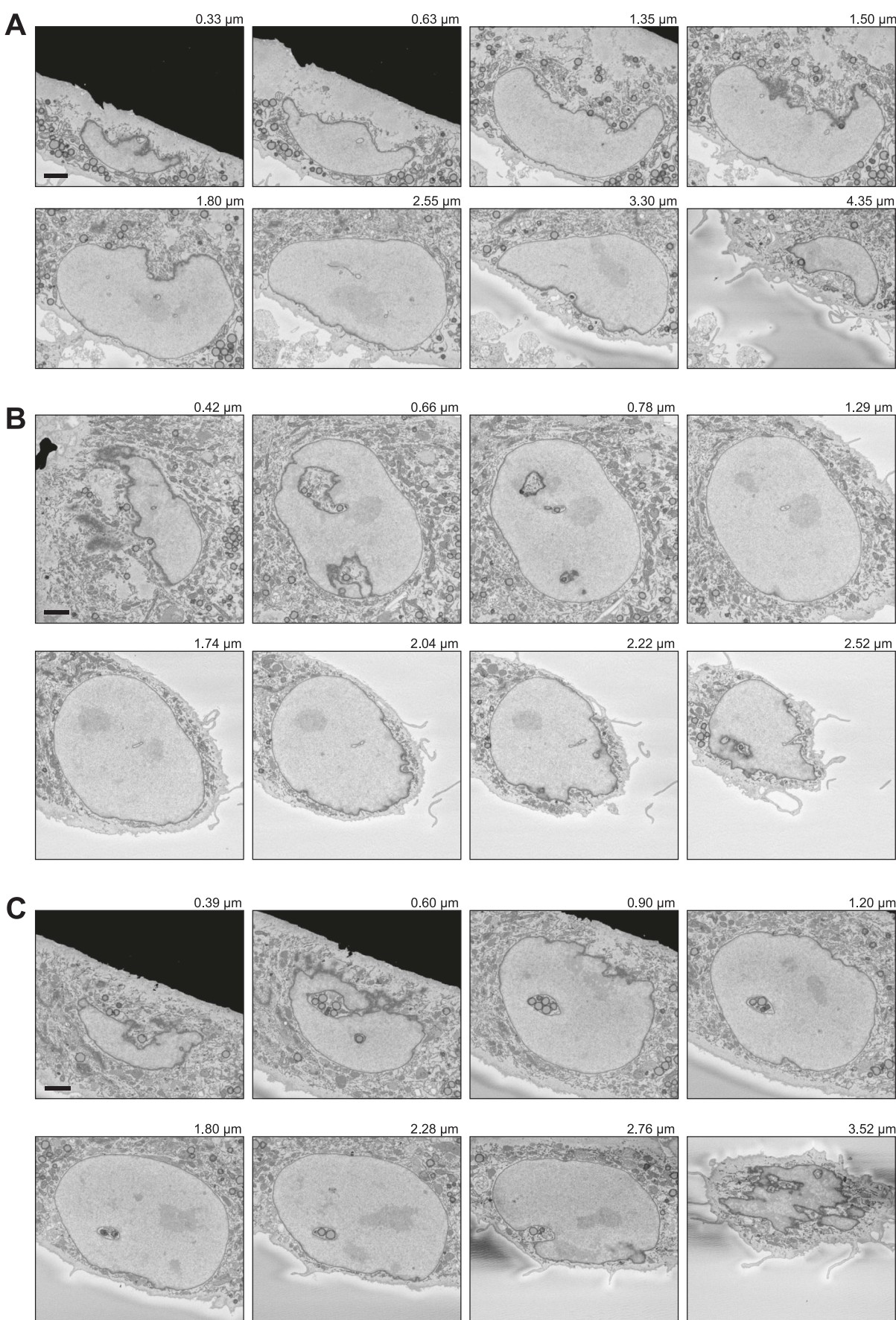

**Figure EV3.  Volume-EM images of human primary macrophages.**

(**A–C**) Cross-sections from SBF-SEM imaging of three individual human primary macrophages, loaded with chol and OA for 2 h, covering the nuclear part of the cells. Cells were imaged using CLvEM and correspond to the images shown in Fig. 4. Numbers above the tiles indicate the distance from the bottom of the corresponding nucleus. Scale bar: 2 µm. Source data are available online for this figure.

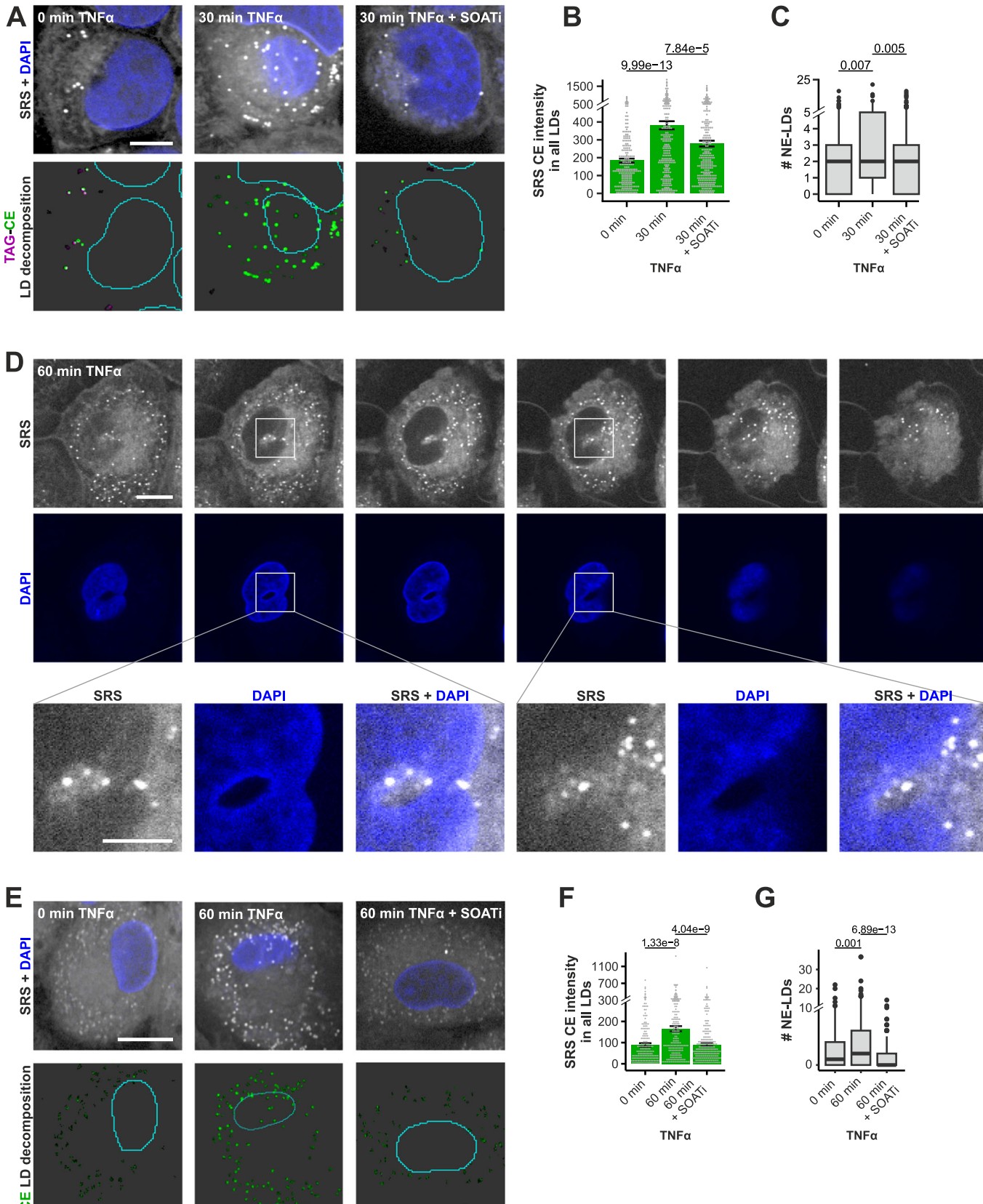

**Figure EV4.  TNFα-induced NE-LDs and their SOAT sensitivity.**

(A) Exemplary SRS images and signal decompositions of A431 cells at 0 and 30 min TNFα treatment ± SOAT inhibition. (B) Quantification of SRS CE intensity in LDs upon treatments as in (A), data presented as mean ± SEM per cell. p values from t-tests. (C) Number of NE-LDs upon treatments as in (A), p values from Mann–Whitney U-tests, boxplots according to ggplot (see Methods). (D) An exemplary z-stack of SRS and DAPI signals at 60 min TNFα treatment showing an NE tunnel containing LDs in a human primary macrophage. (E) Exemplary SRS images and signal decompositions of human primary macrophages at 0 and 60 min TNFα treatment ± SOAT inhibition. (F) Quantification of SRS CE intensity in LDs upon treatments as in (E), data presented as mean ± SEM per cell. p values from t-tests. (G) Number of NE-LDs upon treatments as in (E), p values from Mann–Whitney U-tests, boxplots according to ggplot (see Methods). Cell numbers: 228, 270, 315 for 0, 30, 30 min TNFα + SOATi in (B, C); 225, 210, 278 for 0, 60, 60 min TNFα + SOATi in (F, G). Scale bars: 10 µm (A, D, E) and 5 µm (D inset), distance of consecutive z-slices 0.75 µm. Source data are available online for this figure.

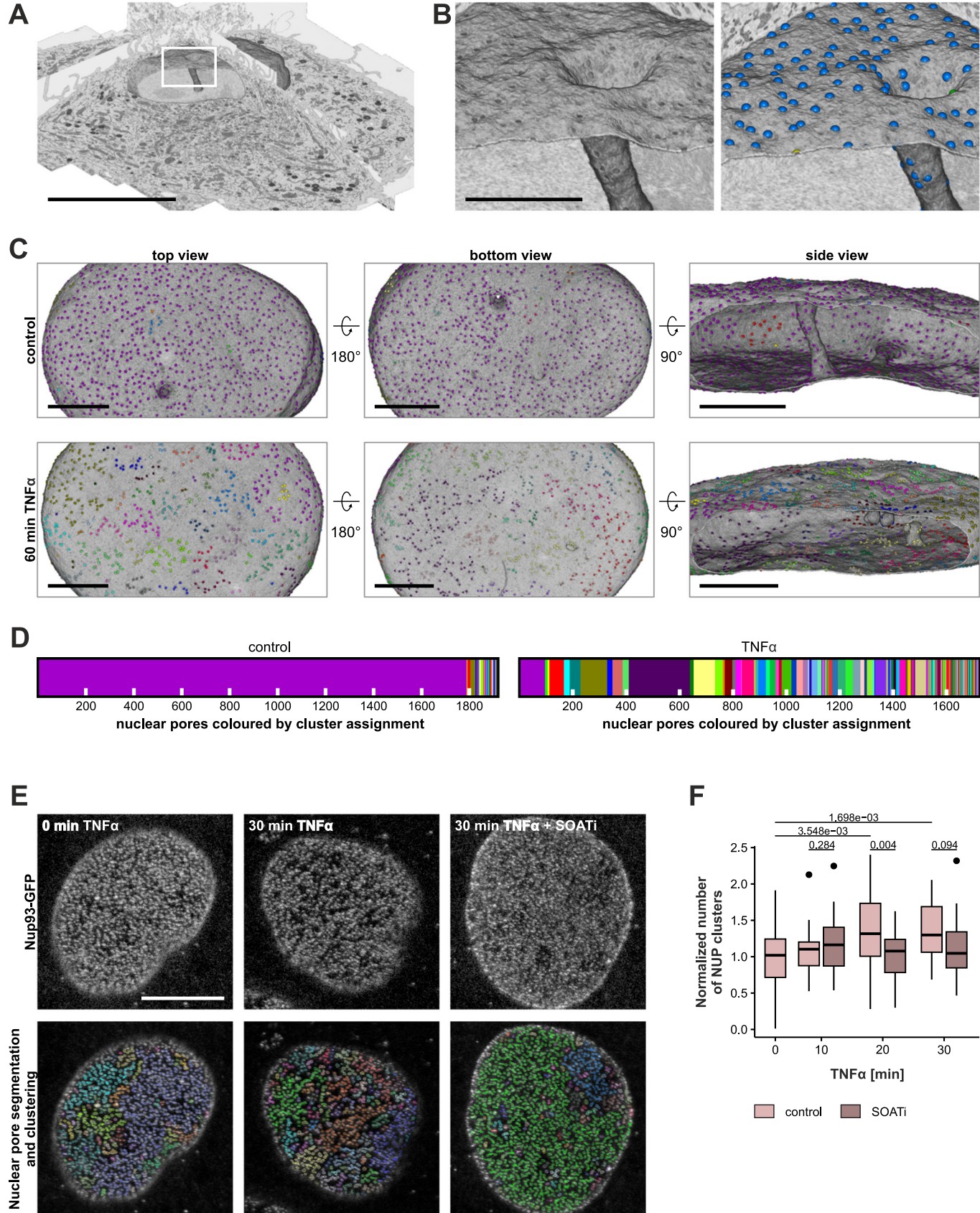

**Figure EV5. Clustering of nuclear pores in TNFα-treated cells.**

(A) 3D visualization of the whole control primary human macrophage cell imaged using FIB-SEM. The image combines orthoslices showing cross-sections from different planes and the volume rendering of the nuclear envelope. (B) Insets from (A) showing the magnified view of the nuclear envelope (left) and the model of the detected nuclear pores (right). (C) Analysis of spatial clustering of the nuclear pores in control and 1 h TNFα-treated macrophages. Pores belonging to the same cluster are visualized with the same color on the top of the 3D volume rendering of the nuclear envelope. The images show the bottom and the side views of the nuclei with the model of the detected nuclear pores. The NEs are clipped at the side views to expose the internal part of the nucleus. (D) Visualization of the number and size of nuclear pore clusters. Each colored stripe indicates one cluster, and its width shows the number of nuclear pores belonging to the cluster. (E) Images of Nup93-GFP expressing A431 cells treated with TNFα ± SOATi as indicated. The lower row shows the segmentation of nuclear pores and contours colored by cluster assignment. (F) Number of clusters normalized to 0 min treatment in A431 cells under 0, 10, 20, and 30 min of TNFα treatment ± SOATi. Cell numbers: 37, 31, 27, 38, 30, 28, and 24 from left to right, data pooled from two biological replicates. $p$ values from $t$-tests and boxplots according to ggplot (see Methods). Scale bars: 10 μm (A, E), 2 μm (B), 3 μm (C). Source data are available online for this figure.

