## [Peer Review File · The EMBO Journal]

Nuclear-envelope-associated lipid droplets are enriched in cholesteryl esters and increase during inflammatory signaling

Ábel Szkalistry, Lauri Vanharanta, Hodaka Saito, Csaba Vörös, Shiqian Li, Antti Isomäki, Teemu Tomberg, Clare Strachan, Ilya Belevich, Eija Jokitalo, and Elina Ikonen

Corresponding author(s): Elina Ikonen (elina.ikonen@helsinki.fi), Lauri Vanharanta (lauri.vanharanta@helsinki.fi), Ábel Szkalistry (szkabel94@gmail.com)

Review Timeline:

Submission Date:	3rd Sep 24
Editorial Decision:	21st Oct 24
Revision Received:	27th Jan 25
Editorial Decision:	19th Feb 25
Revision Received:	4th Mar 25
Accepted:	11th Mar 25

Editor: William Teale

Transaction Report:

Dear Elina,

Thank you again for the submission of your manuscript entitled "Nuclear envelope associated lipid droplets are enriched in cholesteryl esters and increase during inflammatory signaling" and for your patience during the review process. We have now received the reports from three referees, which I copy below.

As you can see from their comments, the referees are in broad agreement that there is room for the manuscript to be strengthened in various directions. That said, all of them point out that the work is timely and may represent an important technological advance.

Based on the overall interest expressed in the reports, I would like to invite you to address the comments of all referees in a revised version of the manuscript. This version should concentrate on addressing the referees' concerns about characterising the role of SOAT1 and how it relates to CE-rich lipid droplets in a more physiological setting. I should add that it is The EMBO Journal policy to allow only a single major round of revision and that it is therefore important to resolve the main concerns at this stage. I believe the concerns of the referees are reasonable and addressable, but please contact me if you have any questions, need further input on the referee comments or if you anticipate any problems in addressing any of their points. I am always available to discuss these points over Zoom. Please, follow the instructions below when preparing your manuscript for resubmission.

I would also like to point out that as a matter of policy, competing manuscripts published during this period will not be taken into consideration in our assessment of the novelty presented by your study ("scooping" protection). We have extended this 'scooping protection policy' beyond the usual 3 month revision timeline to cover the period required for a full revision to address the essential experimental issues. Please contact me if you see a paper with related content published elsewhere to discuss the appropriate course of action.

Again, please contact me at any time during revision if you need any help or have further questions.

Thank you very much again for the opportunity to consider your work for publication. I look forward to your revision.

Best regards,

William

William Teale, Ph.D.
Editor
The EMBO Journal

When submitting your revised manuscript, please carefully review the instructions below and include the following items:

- 1) a .docx formatted version of the manuscript text (including legends for main figures, EV figures and tables). Please make sure that the changes are highlighted to be clearly visible.
- 2) individual production quality figure files as .eps, .tif, .jpg (one file per figure).
- 3) a .docx formatted letter INCLUDING the reviewers' reports and your detailed point-by-point response to their comments. As part of the EMBO Press transparent editorial process, the point-by-point response is part of the Review Process File (RPF), which will be published alongside your paper.
- 4) a complete author checklist, which you can download from our author guidelines ([https://wol-prod-cdn.literatumonline.com/pb-assets/embo-site/Author Checklist%20-%20EMBO%20J-1561436015657.xlsx](https://wol-prod-cdn.literatumonline.com/pb-assets/embo-site/Author%20Checklist%20-%20EMBO%20J-1561436015657.xlsx)). Please insert information in the checklist that is also reflected in the manuscript. The completed author checklist will also be part of the RPF.
- 5) Please note that all corresponding authors are required to supply an ORCID ID for their name upon submission of a revised manuscript.
- 6) We require a 'Data Availability' section after the Materials and Methods. Before submitting your revision, primary datasets

produced in this study need to be deposited in an appropriate public database, and the accession numbers and database listed under 'Data Availability'. Please remember to provide a reviewer password if the datasets are not yet public (see <https://www.embopress.org/page/journal/14602075/authorguide#datadeposition>). If no data deposition in external databases is needed for this paper, please then state in this section: This study includes no data deposited in external repositories. Note that the Data Availability Section is restricted to new primary data that are part of this study.

Note - All links should resolve to a page where the data can be accessed.

8) For data quantification: please specify the name of the statistical test used to generate error bars and P values, the number (n) of independent experiments (specify technical or biological replicates) underlying each data point and the test used to calculate p-values in each figure legend. The figure legends should contain a basic description of n, P and the test applied. Graphs must include a description of the bars and the error bars (s.d., s.e.m.).

9) We would also encourage you to include the source data for figure panels that show essential data. Numerical data can be provided as individual .xls or .csv files (including a tab describing the data). For 'blots' or microscopy, uncropped images should be submitted (using a zip archive or a single pdf per main figure if multiple images need to be supplied for one panel). Additional information on source data and instruction on how to label the files are available at .

10) We replaced Supplementary Information with Expanded View (EV) Figures and Tables that are collapsible/expandable online (see examples in <https://www.embopress.org/doi/10.15252/embj.201695874>). A maximum of 5 EV Figures can be typeset. EV Figures should be cited as 'Figure EV1, Figure EV2" etc. in the text and their respective legends should be included in the main text after the legends of regular figures.

12) Our journal encourages inclusion of *data citations in the reference list* to directly cite datasets that were re-used and obtained from public databases. Data citations in the article text are distinct from normal bibliographical citations and should directly link to the database records from which the data can be accessed. In the main text, data citations are formatted as follows: "Data ref: Smith et al, 2001" or "Data ref: NCBI Sequence Read Archive PRJNA342805, 2017". In the Reference list, data citations must be labeled with "[DATASET]". A data reference must provide the database name, accession number/identifiers and a resolvable link to the landing page from which the data can be accessed at the end of the reference. Further instructions are available at .

13) In order to increase the reproducibility and reach of your work, The EMBO Journal includes a table of reagents that were used in the study. Please provide this along with your revisions.

Further instructions for preparing your revised manuscript:

- a point-by-point response to the referees' comments, with a detailed description of the changes made (as a word file).
- a word file of the manuscript text.
- individual production quality figure files (one file per figure)
- a complete author checklist, which you can download from our author guidelines (<https://www.embopress.org/page/journal/14602075/authorguide>).

- Expanded View files (replacing Supplementary Information)

We realize that it is difficult to revise to a specific deadline. In the interest of protecting the conceptual advance provided by the work, we recommend a revision within 3 months (19th Jan 2025). Please discuss the revision progress ahead of this time with the editor if you require more time to complete the revisions. Use the link below to submit your revision:

Referee #1:

The manuscript from Szkalicity et al. report a technology-driven interrogation of how lipid droplet composition and location change and impact the function of the nuclear envelope. The authors show that droplets rich in cholesterol ester accumulate near the nuclear envelope, which is affected by cholesterol esterification and inflammatory signaling. Additionally, altering the formation of cholesterol ester impacts p65 (and other proteins) translocation into the nucleus potentially through alterations in membrane pore arrangement. Overall, these studies provide novel insights into the heterogeneity of lipid droplets and how these spatial and compositional changes impact nuclear signaling.

Major

- While the trapping studies show that SOAT1 on the NE can drive NE-associated CE-rich LDs, this approach is somewhat artificial and doesn't tell us what is happening under normal physiological conditions. It would be supportive to show SOAT1 trafficking to the ONM or LDs under cholesterol loading, TNFa, etc. or what would happen if SOAT1 is trapped extranuclear (or at least excluded from the nucleus).
- One potential confounding issue with SOAT1 manipulation is that it will also impact free cholesterol levels and trafficking, which can have a substantial impact on many of the pathways/processing measured in this manuscript. I would encourage the authors to image cholesterol trafficking (e.g., BODIPY-cholesterol) and manipulate SOAT along with inhibitors to cholesterol synthesis to dissociate cholesterol effects from those of CE.
- The Raman detection system is insufficiently described and as such makes it challenging to compare or validate these results with other work. Is multichannel lock-in detection used? What is the sensitivity? The authors should present a control study showing signal magnitude as a function of concentration for a standard sample. This is particularly concerning as "saturation" is mentioned several times in the manuscript, which is very unusual for SRS measurements.
- The power values are not useful without understanding the laser spot sizes. Presenting the information as a flux is much more useful in comparison to other measurements.
- Insufficient information is provided regarding the Raman spectral analysis and background subtraction. At the very least, the SI should contain all background spectra used, a range of representative sample spectra from different areas, ranges of weighting coefficients for the linear combinations, and error bars for these values.

Minor

- Raman spectra presented in Figure 1C are confusing, particularly concerning the x axis. Does the x axis show two different spectral regions? Why was a linear x axis not used? The axis should be labeled "Raman shift (cm⁻¹)".
- Similarly, the y axis on the spectra is confusing. Are these Raman gain values?
- Why was a non-negative linear combination spectral analysis method chosen? What are the relevant weightings for different cellular regions? Why did the authors not use a more standard approach such as PCA or MCR-ALS?
- Please refer to NF-κB trafficking as p65 since this is the component of the complex being measured.
- The effects of manipulating CE metabolism on p65 are intriguing. From the data in hand, can you correlate NE-associated LDs (or CE-enriched ones) with rates of p65 (or other proteins) translocation?
- Suggestions: Several paragraphs end with a sentence referring to a finding, but don't summarize the main findings of that section. I would have with readability to provide a "take home" sentence after each results section.
- Define KASH and a brief rationale for why it is used.
- Please list the dose of methyl-beta-cyclodextrin used. This compound can have off-target effects (i.e., AMPK activation).
- Figure 3E stats???

Referee #2:

LDs are storage organelles filled by a neutral lipid core made of primarily triacylglycerol (TAG) and cholesteryl ester CE. In recent years there has been great advances in understanding LD biology and physiology. However, how the behaviour and function of LDs is influenced by their neutral lipid content remains poor understood mainly due to the lack of approaches to monitor individual lipids in situ.

This manuscript by Szkalisity, Vanharanta and colleagues bring an important solution to this problem by employing hyperspectral stimulated Raman scattering microscopy, a methodology allowing the authors to spatially resolve TAG and CE and obtain quantitative information on the relative amounts of these neutral lipids. Not being an expert in SRS microscopy, I am not competent to comment on the specific technical details of spectral decomposition. However, I was convinced that the authors can discriminate between TAG and CE and consider this an important advance.

The authors subsequently used this approach to characterize CE-enriched LDs. They show that these LDs are not uniformly distributed throughout the cytoplasm but concentrate in the proximity of the nuclear envelope. This was observed in multiple cell lines including macrophages, a cell type that known to accumulate CE-rich LDs. This distribution of CE-rich LDs is consistent with the enrichment of SOAT, the CE-synthesizing enzyme, in the nuclear envelope, as previously reported by the same authors. The concentration of CE-rich LDs at the nuclear envelope was increased when SOAT1 was artificially trapped at this region using an elegant strategy.

Using SBF-SEM, it is shown that CE-rich LDs are frequently in proximity of nuclear envelope invaginations. Similar membrane invaginations were observed in cells with low levels of CE-rich LDs (macrophages not loaded with cholesterol) and the significance of the proximity between CE-rich LDs and membrane invaginations remains unclear.

In an attempt to find physiological signals that stimulate CE-rich LD formation, the authors stimulated cells with TNFα, known to activate sphingomyelinase and result in the mobilization of plasma membrane cholesterol and their availability for esterification by SOAT1. Consistent with this idea, TNFα treatment led to an increase in LDs. They further tested if nuclear translocation of the TNFα effector NF-κB and downstream signalling (such as expression of IL-1β) were affected by manipulation of SOAT1 activity.

Overall, this an interesting manuscript that brings an important new tool for studying LDs, with the main focus on their neutral lipid content. While this new message should in principle be of interest to the broad readership of EMBO journal, there are a number of issues that should be clarified/resolved. In particular, it would be useful to understand the relationship between SOAT1 (its distribution, levels, activity) and the spatial distribution of CE-rich LDs. Insights into the relation between TAG and CE synthesizing enzymes and how their distribution affect the relative LD content would also be informative. The links between CE-rich LDs, TNFα signalling and inflammation are exciting but not well developed and in their current form it is unclear if they provide a meaningful addition to the study.

Main points:

1- The writing of the manuscript is not always clear and many critical points are not properly explained for a broad audience.

Some examples are listed below.

- a. Fig 1B: some cellular structures (mitochondria? ER? Other membranes?) are detected and it may be appropriate to explain from where these are coming from and how the authors deal with that.
- b. Fig 1C: it is not explained how the SRS image on the left and the graphs on the right relate to each other.
- c. It is stated that TAG and CE peaks "results from the symmetric CH₂ stretching in the open fatty acid chains while the latter results from the symmetric CH₂ stretching within the sterol ring". But it would be good to mention how these CH₂ can be distinguished from the ones in fatty acids from phospholipids and in the rings of non-esterified sterols, respectively
- d. "The weights from the linear combination for TAG and CE references were considered as direct estimates of TAG and CE lipid amounts. For visualization the weights were scaled and back-projected to the SRS images to create false-colored images of TAGs and CEs with magenta and green". More detail on this would also be useful.
- e. what is nuclear reticulum? It's not explained or referenced.

2- The authors attempt to establish a correlation between the previously reported distribution of SOAT1 and LDs enriched in CE.

This is supported by the SOAT1 trapping experiment. However, a more in-depth analysis of the localization of endogenous SOAT1 and DGATs in the various cell lines and the conditions tested would be beneficial. This analysis might help explain why the correlation between CE enrichment in LDs and their proximity to the nuclear envelope is not consistent across all cells. While very clear in A431 cells, it is less obvious in U2OS and macrophages (Figure 3D,E). Based on TAG-CE decomposition color code, most LDs in U2OS cells appear strongly enriched in CE. On the other hand, in macrophages, Chol+OA loading appears to result in CE accumulation throughout with TAG being less abundant in a population of LDs, including some in proximity of the NE. It is also ignored most of the LDs in these cells appear to be peripheral and according to Fig 3D contain significant levels of CE. The analysis of the distribution and levels of SOAT1 and DGATs may also clarify some intriguing observations on the relative CE and TAG levels in several experiments. For example, in KGN cells (Fig 2B) loading of OA or Chol alone stimulates TAG and CE synthesis as expected. However, in cells loaded simultaneously with Chol+OA, there is a strong boost of CE synthesis and only basal TAG levels. In addition, in the SOAT1 trapping experiments the amount of TAG was lower in the NE-trapped cells compared to the non-trapped expressing cells, as pointed out by the authors.

3- The induction of CE-rich LDs by TNF α treatment is interesting. However, the role of CE in facilitating the nuclear accumulation of NF-KB and a soluble fluorescent reporter requires further characterization. As it stands, the experiments are difficult to interpret. Similarly, the effect of SOAT1 inhibition in dampening the NF-KB nuclear translocation and signalling is potentially interesting. But again, these studies are superficial and the evidence for a direct effect of CE and its accumulation in NE-LDs proximal to the nucleus isn't strong. How these observations relate to the changes in nuclear pore distribution remains unclear.

Minor points:

- Authors refer to NE-LDs vs cytoplasmic LDs. This terminology can be misleading as all LDs studied in this manuscript including NE-LDs are cytoplasmic.

Referee #3:

In this manuscript, the authors introduce their approach to using stimulated Raman scattering (SRS) microscopy to distinguish between triglycerides (TAGs) and cholesteryl esters (CEs) in mammalian cells. While the application of SRS for imaging TAGs versus CEs has been previously reported (PMID: 24869754), the decomposition method utilized in this study offers more quantitative measurements. The authors further demonstrate the presence of lipid droplets (LDs) associated with the nuclear envelope, which appear to be enriched in CEs. When SOAT1, the enzyme responsible for cholesterol ester synthesis, is artificially tethered to the nuclear envelope, an increase in nuclear envelope-associated LDs is observed. The authors also show that TNF- α treatment induces NF- κ B nuclear translocation, a process enhanced by SOAT1 overexpression and suppressed by SOAT1 inhibition. Moreover, manipulation of SOAT1 alters the nuclear translocation of fluorescent Dendra, suggesting a broader role for CE levels in regulating nuclear transport. The study concludes by extending these findings to human primary macrophages treated with TNF- α , where similar effects on nuclear translocation are observed.

Overall, the manuscript highlights the identification of nuclear envelope-associated LDs and their regulation by SOAT1, as well as the involvement of SOAT1 in modulating NF- κ B nuclear translocation in response to pro-inflammatory signals. However, the relationship between SOAT1's role in regulating nuclear envelope-associated LDs and NF- κ B nuclear translocation remains unclear, leaving a gap in understanding the underlying mechanisms. Additionally, the quality of the presented data needs to be improved to fully support the authors' conclusions. In its current form, the manuscript requires substantial revision before it can be considered suitable for publication.

Major points:

1. Statistical analyses are required for all the data shown in the manuscript to validate the conclusions drawn from the studies. Additionally, for all the bar graphs presented, please include individual data points. This will allow the reader to evaluate the variability and distribution of the data. It is also important to specify the number of replicates and clarify whether these are technical or biological replicates.
2. Changes in nuclear envelope-associated LDs under different conditions are central claims of the manuscript. Since overall LD numbers may change under different conditions, it is crucial to normalize these results, similar to the approach used in Figure 5G. This normalization should be applied to Figures 6C and 7B to ensure the validity of the conclusions. In many images, the selected cells that display increased nuclear envelope-associated LDs also appear to have a higher overall number of LDs compared to neighboring cells.
3. To better visualize nuclear envelope-associated LDs, it is important to include DAPI images in addition to the outlined regions

shown in Figures 3D, 6B, and 7A. This will provide more accurate localization of the nuclear envelope. In particular, for Figure 7A, the outline seems to be positioned outside the nucleus, which raises concerns about the accuracy of LD localization.

4. In Figure 5D, the nuclear shape appears altered due to the artificial tethering of SOAT1 to the nucleus. Are the increased nuclear envelope-associated LDs associated with these changes in nuclear shape? Additionally, under this condition, the nuclear envelope-associated LDs are enriched with both CEs and TAGs, which contradicts the authors' claim that "these results argue that the NE-localization of SOAT1 assists in the generation of CE-rich LDs in the NE." The authors should provide a clearer explanation for this discrepancy.

5. To support the conclusion that SOAT1 inactivation or overexpression affects NF- κ B nuclear translocation, the authors should present data for the entire time course (e.g., over 30 minutes), as shown in Figure 6F. Comparing the nuclear translocation at only the 20-minute mark is not sufficient, especially since the observed differences are quite subtle.

6. To clarify the relationship between SOAT1's role in regulating nuclear envelope-associated LDs and NF- κ B nuclear translocation, the authors should not limit their analysis to SOAT1 overexpression. They should also assess the effects of SOAT1 tethering to the nucleus. Does SOAT1 tethering have a stronger effect on promoting NF- κ B nuclear translocation?

7. The authors should include SBF-SEM data for control cells to allow for direct comparison and validation of the findings.

Minor Points:

In the sentence "We next want to understand if this acute cholesterol redistribution...", what evidence supports the claim that this redistribution is acute? The authors should provide more information to justify this characterization.

EMBOJ-2024-118923

Szkalitsy, Vanharanta et al. Nuclear envelope associated lipid droplets are enriched in cholesteryl esters and increase during inflammatory signaling

Point-by-point responses to the Reviewers' comments

Referee #1:

The manuscript from Szkalitsy et al. report a technology-driven interrogation of how lipid droplet composition and location change and impact the function of the nuclear envelope. The authors show that droplets rich in cholesterol ester accumulate near the nuclear envelope, which is affected by cholesterol esterification and inflammatory signaling. Additionally, altering the formation of cholesterol ester impacts p65 (and other proteins) translocation into the nucleus potentially through alterations in membrane pore arrangement. Overall, these studies provide novel insights into the heterogeneity of lipid droplets and how these spatial and compositional changes impact nuclear signaling.

We thank the Reviewer for the careful assessment, positive remarks and valuable feedback on our study. In the revised manuscript, we have addressed the specific points raised by the Reviewer, as detailed below.

Major

#1/1) While the trapping studies show that SOAT1 on the NE can drive NE-associated CE-rich LDs, this approach is somewhat artificial and doesn't tell us what is happening under normal physiological conditions. It would be supportive to show SOAT1 trafficking to the ONM or LDs under cholesterol loading, TNF α , etc. or what would happen if SOAT1 is trapped extranuclear (or at least excluded from the nucleus).

Thank you for raising this important point. We agree that studying the trafficking and distribution of SOAT1 under physiological conditions can add supportive information. We therefore generated cells where SOAT1 is endogenously SNAP-tagged and visualized its distribution and trafficking in TNF α stimulated cells (we considered that pharmacological cholesterol loading from cyclodextrin is less physiological and is known to perturb the architecture of the ER). This revealed that TNF α induces a change in SOAT1 localization with an overall redistribution of the enzyme towards the perinuclear region (for comparison, the distribution of endogenous DGAT1 was not altered as judged by immunostaining). In addition, we observed SOAT1 foci moving dynamically in the ER in the proximity of the CE-enriched LDs forming in the outer nuclear membrane. This matches well with the observation that under these conditions, CE-rich LDs are generated at the NE. These data are included in the new **Fig 6E-G** and **Fig S2**. We have so far not found any physiological conditions where SOAT1 would be excluded from the nuclear envelope.

To further substantiate the physiological role of SOAT1 in generating NE-LDs, we have also included data showing that acute pharmacological SOAT1 inhibition inhibits the

formation of $\text{TNF}\alpha$ induced NE-associated CE-rich LDs in two cell systems (A431 cells in **Fig EV4A-C** and human primary monocyte-macrophages in **Fig EV4E-G**).

#1/2) One potential confounding issue with SOAT1 manipulation is that it will also impact free cholesterol levels and trafficking, which can have a substantial impact on many of the pathways/processing measured in this manuscript. I would encourage the authors to image cholesterol trafficking (e.g., BODIPY-cholesterol) and manipulate SOAT along with inhibitors to cholesterol synthesis to dissociate cholesterol effects from those of CE.

We agree with the Reviewer that SOAT1 manipulations affect the free vs. esterified cholesterol balance and that it is very challenging (if not impossible) to rule out a potential role of altered free cholesterol levels in the processes under study. We have therefore now included this aspect in the Discussion (beginning of p. 12). Interestingly, we did find that SOAT1 overexpression does not significantly affect total cellular free cholesterol levels and have now added this information in the Results (bottom of p. 7). Nevertheless, this does not exclude the possibility of local changes in the cholesterol content of subcellular membrane domains. We did not use BODIPY-cholesterol in this case, as we have previously found that it is poorly esterified by SOAT1 (Hölttä et al. Traffic 2008, PMID: 18647169), and we also felt that using inhibitors of cholesterol synthesis in the conditions under study (in the presence of complete medium, i.e. serum lipoproteins, where endogenous cholesterol synthesis is low) will not conclusively dissociate the effects between free and esterified cholesterol.

#1/3) The Raman detection system is insufficiently described and as such makes it challenging to compare or validate these results with other work. Is multichannel lock-in detection used? What is the sensitivity? The authors should present a control study showing signal magnitude as a function of concentration for a standard sample. This is particularly concerning as "saturation" is mentioned several times in the manuscript, which is very unusual for SRS measurements.

Thank you for these valid comments. The Raman detection system is now thoroughly described in a separate study that we have recently published (Tomberg et al. Optics Continuum, 2024, doi.org/10.1364/OPTCON.532676) and that we refer to in the revised manuscript. Specifically, no multichannel lock-in, just a single-channel lock-in detection was used. As requested, we have also carried out control experiments demonstrating how the signal magnitude changes as a function of the concentration of a standard sample (see **Figure 1 for reviewer** below). We wish to note that we referred to the lock-in detector signal saturation and not to the vibrational transition and have now specified this in the Methods (p. 16). As such, lock-in signal saturation due to a strong SRS signal is not an unusual phenomenon.

Figure 1 for Reviewer. SRS microscope sensitivity for a standard sample. Correlation between known TAG concentration and measured SRS intensity. Triolein was diluted in chloroform:methanol (2:1) to 1,10,50,100 mM concentration and imaged with the SRS settings used in the study. SRS intensity is defined as the TAG component weight after the linear decomposition of the signal to TAG and solvent components using the solvent-silent ($1625\text{-}1775\text{ cm}^{-1}$) region.

#1/4) The power values are not useful without understanding the laser spot sizes. Presenting the information as a flux is much more useful in comparison to other measurements.

We have now presented the power values as fluxes in **Appendix Table S1**.

#1/5) Insufficient information is provided regarding the Raman spectral analysis and background subtraction. At the very least, the SI should contain all background spectra used, a range of representative sample spectra from different areas, ranges of weighting coefficients for the linear combinations, and error bars for these values.

As requested, we have added a more thorough description of the Raman spectral analysis and background subtraction (Methods p. 15). As also requested, all the background spectra used are now available in the source data files associated with the false-colored decomposition images. Additionally, we have included a range of representative sample spectra from different areas, as well as representative ranges of weighting coefficients for the linear combinations and their variation (**Fig EV1A-C**).

Minor

#1/6) Raman spectra presented in Figure 1C are confusing, particularly concerning the x axis. Does the x axis show two different spectral regions? Why was a linear x axis not used? The axis should be labeled "Raman shift (cm⁻¹)".

We apologize for the unclear presentation. We have now improved the presentation in the x-axis of **Fig 1C**, as suggested. Indeed, the x axis show two different spectral

regions, which is the reason for the cut in the x axis. The separate ranges are equally scaled.

#1/7) Similarly, the y axis on the spectra is confusing. Are these Raman gain values?

The y axis shows stimulated Raman loss. This is now indicated in the legend of **Fig 1C**.

#1/8) Why was a non-negative linear combination spectral analysis method chosen? What are the relevant weightings for different cellular regions? Why did the authors not use a more standard approach such as PCA or MCR-ALS?

Thank you for this remark. We initially tested several unsupervised techniques for SRS spectral decomposition including PCA, MCR-ALS, and UMAP. Both UMAP and PCA were able to separate the pixels in the different loads at high signal intensities. However, at low signal intensities the pixels of different loads were heavily mixed, as shown in the **Figure 2 for Reviewer** below. Additionally, as in those techniques the reference spectrum is identified by the model, its identity needs to be separately determined. Therefore, the vanilla versions of the mentioned techniques are less appropriate for the task of distinguishing CEs and TAGs than our direct approach. Notably, our model can be considered as a modification of the constrained MCR-ALS with “hard-wired” reference spectra instead of soft constraints. It would be worthwhile testing if a soft-constrained MCR-ALS is performing better than our approach, but a comprehensive comparison of signal decomposition techniques for our task is beyond the scope of the present study, and the biochemical TLC-based verifications showed that our approach is working reasonably well.

Figure 2 for Reviewer. Unsupervised visualization of SRS spectra with UMAP and PCA. (a) Unbiased view of pixel-level LD SRS signals from lipid starved or loaded cells as indicated, visualized with uniform manifold approximation and projection (UMAP) at pixel level. **(b)** Overlaying average signal intensity to the panel in (a), excluding outliers. For lower signal intensities the pixels of different loads were heavily mixed (top-left arm). For higher signal intensities (right arm and c), the UMAP plot was able to separate the pixels coming from different loads. **(c)** Inset of the indicated area in (a). **(d)** PCA plot of SRS signals for principal components (PC) 1-2 (top) and 3-4 (bottom) for the same data as in (a). **(e)** Overlaying average signal intensity to panels in (d). PC1 captures variance in signal intensity, but only PC3 and PC4 separate the loads.

#1/9) Please refer to NF- κ B trafficking as p65 since this is the component of the complex being measured.

We have now changed the nomenclature, as suggested.

#1/10) The effects of manipulating CE metabolism on p65 are intriguing. From the data in hand, can you correlate NE-associated LDs (or CE-enriched ones) with rates of p65 (or other proteins) translocation?

This is an interesting question. Unfortunately, these parameters cannot be correlated in the same cells, because the immunodetection of p65 (or other proteins) requires detergent or solvent permeabilization of membranes, which pending on the method diminishes or abolishes lipid storage in droplets. What we do observe, though, is that TNF α treatment increases NE-associated CE-LDs and that this correlates with enhanced p65 nuclear translocation in similarly treated cells.

#1/11) Suggestions: Several paragraphs end with a sentence referring to a finding, but don't summarize the main findings of that section. I would have with readability to provide a "take home" sentence after each results section.

Thank you for this good suggestion. We have now included such concluding sentences in the Results section.

#1/12) Define KASH and a brief rationale for why it is used.

KASH stands for Klarsicht, ANC-1 and Syne homology), a protein in the outer nuclear membrane. Its KASH peptide locates in the perinuclear space where it binds the SUN (Sad1, UNC-84) protein anchored to the inner nuclear membrane, with SUN-KASH complexes forming bridges across the perinuclear space. When SOAT is fused to KASH peptide, its binding to SUN anchors the fusion protein to the NE (please see schematic cartoon in revised **Fig 5B** and reference Sosa *et al*, 2012 PMID: 22632968).

#1/13) Please list the dose of methyl-beta-cyclodextrin used. This compound can have off-target effects (i.e., AMPK activation).

We have now provided this information in the Methods (p. 13).

#1/14) Figure 3E stats???

The statistics of **Fig 3E** were included on the following page (now that the layout has changed in the revised manuscript, this is no longer a problem).

Referee #2:

LDs are storage organelles filled by a neutral lipid core made of primarily triacylglycerol (TAG) and cholesteryl ester CE. In recent years there has been great advances in understanding LD biology and physiology. However, how the behaviour and function of LDs is influenced by their neutral lipid content remains poorly understood mainly due to the lack of approaches to monitor individual lipids in situ.

This manuscript by Szkalitsy, Vanharanta and colleagues bring an important solution to this problem by employing hyperspectral stimulated Raman scattering microscopy, a methodology allowing the authors to spatially resolve TAG and CE and obtain quantitative information on the relative amounts of these neutral lipids. Not being an expert in SRS microscopy, I am not competent to comment on the specific technical details of spectral decomposition. However, I was convinced that the authors can discriminate between TAG and CE and consider this an important advance.

The authors subsequently used this approach to characterize CE-enriched LDs. They show that these LDs are not uniformly distributed throughout the cytoplasm but concentrate in the proximity of the nuclear envelope. This was observed in multiple cell lines including macrophages, a cell type that is known to accumulate CE-rich LDs. This distribution of CE-rich LDs is consistent with the enrichment of SOAT, the CE-synthesizing enzyme, in the nuclear envelope, as previously reported by the same authors. The concentration of CE-rich LDs at the nuclear envelope was increased when SOAT1 was artificially trapped at this region using an elegant strategy.

Using SBF-SEM, it is shown that CE-rich LDs are frequently in proximity of nuclear envelope invaginations. Similar membrane invaginations were observed in cells with low levels of CE-rich LDs (macrophages not loaded with cholesterol) and the significance of the proximity between CE-rich LDs and membrane invaginations remains unclear.

In an attempt to find physiological signals that stimulate CE-rich LD formation, the authors stimulated cells with TNF α , known to activate sphingomyelinase and result in the mobilization of plasma membrane cholesterol and their availability for esterification by SOAT1. Consistent with this idea, TNF α treatment led to an increase in LDs. They further tested if nuclear translocation of the TNF α effector NF- κ B and downstream signalling (such as expression of IL-1 β) were affected by manipulation of SOAT1 activity.

Overall, this is an interesting manuscript that brings an important new tool for studying LDs, with the main focus on their neutral lipid content. While this new message should in principle be of interest to the broad readership of EMBO journal, there are a number of issues that should be clarified/resolved. In particular, it would be useful to understand the relationship between SOAT1 (its distribution, levels, activity) and the spatial distribution of CE-rich LDs. Insights into the relation between TAG and CE synthesizing enzymes and how their distribution affect the relative LD content would also be informative. The links between CE-rich LDs, TNF α signalling and inflammation are exciting but not well developed and in their current form it is unclear if they provide a meaningful addition to the study.

We thank the Reviewer for the thorough evaluation and overall positive remarks regarding our manuscript. We have now carefully considered the comments and

valuable suggestions made. We have added data on the distribution of SOAT1 when it becomes activated upon $\text{TNF}\alpha$ stimulation, including its spatial relationship to CE-rich LDs. The question regarding the interplay between TAG and CE synthesizing enzymes in the ER is very interesting and would deserve a study of its own, especially considering the multiple ubiquitously expressed TAG generating enzymes identified to date. In the present work, we only compared the distribution of endogenous SOAT1 to that of endogenous DGAT1 (for which we found working antibodies and which is mainly responsible for the initial TAG synthesis upon oleic acid loading in A431 cells used in the present study). We agree that the links between CE-rich LDs, $\text{TNF}\alpha$ signalling and inflammation remain open at present, and hope to dissect the precise mechanistic explanations between these observations further in our future studies. The specific points raised by the Reviewer are addressed, as detailed below.

Main points:

#2/1)- The writing of the manuscript is not always clear and many critical points are not properly explained for a broad audience. Some examples are listed below.

#2/1a). Fig 1B: some cellular structures (mitochondria? ER? Other membranes?) are detected and it may be appropriate to explain from where these are coming from and how the authors deal with that.

Indeed, SRS signals recorded in the detected region provide prominent signals not only from lipid droplets but also other lipid-rich structures, especially the plasma membrane and all endomembrane compartments. We have now brought this up in the Results (p. 4).

#2/1b). Fig 1C: it is not explained how the SRS image on the left and the graphs on the right relate to each other.

We thank the Reviewer for noticing this logical inconsistency in the figure. We have now reorganized the panels of Figure 1, moving the false-colored, decomposed images into panel **Fig 1E** and explaining their relationship to the images in Fig 1C.

#2/1c). It is stated that TAG and CE peaks "results from the symmetric CH_2 stretching in the open fatty acid chains while the latter results from the symmetric CH_2 stretching within the sterol ring". But it would be good to mention how these CH_2 can be distinguished from the ones in fatty acids from phospholipids and in the rings of non-esterified sterols, respectively

Thank you for the remark. The origin of the CH_2 groups (e.g. whether they come from the rings of non-esterified or esterified sterols) cannot be distinguished solely based on the highlighted peaks. To distinguish these molecules, one needs to look at the full spectrum and multiple peaks: e.g., if the peak of the open fatty acid chain CH_2 is missing but the sterol CH_2 is present then it is more likely to be a non-esterified sterol rather than CE. We have now clarified this sentence (Results, p. 4).

#2/1d). "The weights from the linear combination for TAG and CE references were considered as direct estimates of TAG and CE lipid amounts. For visualization the weights were scaled and back-projected to the SRS images to create false-colored images of TAGs and CEs with magenta and green". More detail on this would also be useful.

We have now more explicitly explained this, with the help of the revised **Fig EV1A-C** as well as in the Methods (p. 16).

#2/1e). what is nuclear reticulum? It's not explained or referenced.

With this term we actually referred to nucleoplasmic reticulum, we have now reformulated it and added a reference (Malhas *et al*, 2011 PMID: 21514163).

#2/2)- The authors attempt to establish a correlation between the previously reported distribution of SOAT1 and LDs enriched in CE. This is supported by the SOAT1 trapping experiment. However, a more in-depth analysis of the localization of endogenous SOAT1 and DGATs in the various cell lines and the conditions tested would be beneficial. This analysis might help explain why the correlation between CE enrichment in LDs and their proximity to the nuclear envelope is not consistent across all cells. While very clear in A431 cells, it is less obvious in U2OS and macrophages (Figure 3D,E). Based on TAG-CE decomposition color code, most LDs in U2OS cells appear strongly enriched in CE. On the other hand, in macrophages, Chol+OA loading appears to result in CE accumulation throughout with TAG being less abundant in a population of LDs, including some in proximity of the NE. It is also ignored most of the LDs in these cells appear to be peripheral and according to Fig 3D contain significant levels of CE. The analysis of the distribution and levels of SOAT1 and DGATs may also clarify some intriguing observations on the on the relative CE and TAG levels in several experiments. For example, in KGN cells (Fig 2B) loading of OA or Chol alone stimulates TAG and CE synthesis as expected. However, in cells loaded simultaneously with Chol+OA, there is a strong boost of CE synthesis and only basal TAG levels. In addition, in the SOAT1 trapping experiments the amount of TAG was lower in the NE-trapped cells compared to the non-trapped expressing cells, as pointed out by the authors.

Thank you for the careful assessment of the data and for bringing up these important aspects. Indeed, it is possible that the differences in CE and TAG levels between the cell types studied may in part relate to the different levels and/or distributions of the respective enzymes. We agree that especially the localization of endogenous SOAT1 and DGATs in different cell types and under variable physiological conditions is an interesting future question to investigate. In the context of the revision of the present work, we carried out the following experiments:

To study the distribution and dynamics of endogenous SOAT1, we generated cells where SOAT1 is endogenously SNAP-tagged and visualized its distribution and trafficking in TNF α stimulated cells. (Of note, SOAT2 is not expressed in A431 cells, and we have previously shown that a fluorophore-tagged SOAT1 retains activity). This

revealed that SOAT1 is broadly and rather diffusely distributed throughout the ER. TNF α induced a redistribution of the enzyme to the perinuclear region, including the nuclear envelope. Moreover, we found perinuclear SOAT1 clusters to move dynamically in the proximity of the generated CE-rich LDs. This matches well with the observation that under these conditions, CE-rich LDs are found at the nuclear envelope. These data are included in the revised **Fig 6E-G**.

We also scrutinized the relative contributions of DGAT1 and DGAT2 in the generation of TAG LDs in A431 cells. We found that at early times of TAG LD formation, DGAT1 is the major enzyme responsible for TAG synthesis (see **Figure 3 for Reviewer** below). This fits with previous observations in e.g. Huh7 cells, showing DGAT2 activity at later time points of oleic acid loading (ref. PMID: 38809969). We therefore compared the distributions of endogenous SOAT1 and DGAT1 that are structurally closely related enzymes. When assessing the distribution of endogenous DGAT1 by antibody staining upon TNF α treatment, we found that distinctly from SOAT1, DGAT1 was not prominently redistributed to the perinuclear region. These data, now included in the new **Fig S2**, might in part explain the relative TAG-de-enrichment of NE-associated LDs.

Figure 3 for Reviewer. Role of DGAT1&2 in early TAG synthesis. TLC analysis of TAG levels in wild-type A431 cells loaded with 200 μM oleic acid (OA) for 30 min +/- DGAT1 inhibitor or DGAT2 inhibitor as indicated.

Finally, we wish to point out that despite considerable cell-to-cell variation (as also exemplified in the Figures), systematic quantification of SRS signals from hundreds of cells demonstrates a significant correlation between the CE enrichment of LDs and their association with the NE, as demonstrated e.g. in the quantification in **Fig 3E**. Moreover, we emphasize that CEs can also be found in other, non-NE associated LDs that we term cytoplasmic LDs; it is just that CEs are more enriched in the NE-associated LDs compared to the cytoplasmic LDs. We have now better articulated this point and improved the accuracy of presentation in the quantification of the relative CE-enrichment in NE-LDs in the respective Figures. We have also exchanged the exemplary panel of U2OS cells with a more representative one (**Fig 3D**).

#2/3) The induction of CE-rich LDs by TNF α treatment is interesting. However, the role of CE in facilitating the nuclear accumulation of NF-KB and a soluble fluorescent reporter requires further characterization. As it stands, the experiments are difficult to interpret. Similarly, the effect of SOAT1 inhibition in dampening the NF-KB nuclear translocation and signalling is potentially interesting. But again, these studies are superficial and the evidence for a direct effect of CE and its accumulation in NE-LDs proximal to the nucleus isn't strong. How these observations relate to the changes in nuclear pore distribution remains unclear.

We agree with the Reviewer that this is by nature an observational study that builds on the SRS imaging regime established. However, we propose that the findings reported are likely to be, at least in part, causally linked – such as the NE localization of endogenous SOAT1 contributing to the CE-enrichment of NE-associated LDs. To further strengthen the relationship between nuclear translocation, nuclear pore distribution and SOAT1, we have added data showing that the nuclear pore distribution becomes uneven also upon TNF α treatment of A431 cells and that this is linked to SOAT1 activity, as it is abrogated by SOAT1 inhibition (new **Fig EV5E, F**). Despite this, we acknowledge that the dissection of how exactly the altered free cholesterol-cholesteryl ester balance controlled by SOAT1 affects nuclear pore permeability remains to be mechanistically addressed in future work.

Minor points:

#2/4) Authors refer to NE-LDs vs cytoplasmic LDs. This terminology can be misleading as all LDs studied in this manuscript including NE-LDs are cytoplasmic.

We agree and have therefore now more clearly stated that we use this terminology to distinguish between NE- and non-NE associated cytoplasmic LDs (p. 7).

Referee #3:

In this manuscript, the authors introduce their approach to using stimulated Raman scattering (SRS) microscopy to distinguish between triglycerides (TAGs) and cholesteryl esters (CEs) in mammalian cells. While the application of SRS for imaging TAGs versus CEs has been previously reported (PMID: 24869754), the decomposition method utilized in this study offers more quantitative measurements. The authors further demonstrate the presence of lipid droplets (LDs) associated with the nuclear envelope, which appear to be enriched in CEs. When SOAT1, the enzyme responsible for cholesterol ester synthesis, is artificially tethered to the nuclear envelope, an increase in nuclear envelope-associated LDs is observed. The authors also show that TNF-alpha treatment induces NF-kB nuclear translocation, a process enhanced by SOAT1 overexpression and suppressed by SOAT1 inhibition. Moreover, manipulation of SOAT1 alters the nuclear translocation of fluorescent Dendra, suggesting a broader role for CE levels in regulating nuclear transport. The study concludes by extending these findings to human primary macrophages treated with TNF-alpha, where similar effects on nuclear translocation are observed.

Overall, the manuscript highlights the identification of nuclear envelope-associated LDs and their regulation by SOAT1, as well as the involvement of SOAT1 in modulating NF-kB nuclear translocation in response to pro-inflammatory signals. However, the relationship between SOAT1's role in regulating nuclear envelope-associated LDs and NF-kB nuclear translocation remains unclear, leaving a gap in understanding the underlying mechanisms. Additionally, the quality of the presented data needs to be improved to fully support the authors' conclusions. In its current form, the manuscript requires substantial revision before it can be considered suitable for publication.

We thank the Reviewer for the careful evaluation and valuable suggestions on how to further improve our study. We admit that the dissection of how exactly the altered free cholesterol-cholesteryl ester balance controlled by SOAT1 affects nuclear pore permeability remains to be mechanistically solved in future work. To strengthen the relationship between nuclear translocation, nuclear pore distribution and SOAT1 activity, we have now added data showing that the nuclear pore distribution becomes uneven also upon TNF α treatment of A431 cells and that this is linked to SOAT1 activity, as it is abrogated by SOAT1 inhibition. These data are included in the new **Fig EV5E, F**. While the observations leave a gap in our understanding, we hope that sharing these data may stimulate future studies.

Overall, we have now improved the quality of the data presented as proposed by the Reviewer. The specific points raised by the Reviewer are addressed as detailed below.

Major points:

#3/1) Statistical analyses are required for all the data shown in the manuscript to validate the conclusions drawn from the studies. Additionally, for all the bar graphs presented, please include individual data points. This will allow the reader to evaluate

the variability and distribution of the data. It is also important to specify the number of replicates and clarify whether these are technical or biological replicates.

Thank you for bringing up these points. In the revised manuscript, we have included statistical analyses supporting the conclusions as well as individual data points for all the bar graphs (**Fig 1H, 2B, 5H, 6D, 8C, E, EV4B, F**). We have also specified the number of replicates, including information on whether they are technical or biological ones. The replicate information is available in the source data files.

#3/2) Changes in nuclear envelope-associated LDs under different conditions are central claims of the manuscript. Since overall LD numbers may change under different conditions, it is crucial to normalize these results, similar to the approach used in Figure 5G. This normalization should be applied to Figures 6C and 7B to ensure the validity of the conclusions. In many images, the selected cells that display increased nuclear envelope-associated LDs also appear to have a higher overall number of LDs compared to neighboring cells.

This is a good point – the LD numbers vary substantially between cells and conditions. Therefore, as suggested, we have now normalized the results also in the original Figures 6C and 7B (these are revised **Fig 6C** and **Fig 8B**). These data show that the changes in NE-LDs hold also after normalizing to the total cellular LD numbers.

#3/3) To better visualize nuclear envelope-associated LDs, it is important to include DAPI images in addition to the outlined regions shown in Figures 3D, 6B, and 7A. This will provide more accurate localization of the nuclear envelope. In particular, for Figure 7A, the outline seems to be positioned outside the nucleus, which raises concerns about the accuracy of LD localization.

Thank you for this suggestion. We have now included the DAPI channel images for the Figures indicated by the Reviewer (**Fig 3D, Fig 6B** and **Fig 8A**).

#3/4) In Figure 5D, the nuclear shape appears altered due to the artificial tethering of SOAT1 to the nucleus. Are the increased nuclear envelope-associated LDs associated with these changes in nuclear shape? Additionally, under this condition, the nuclear envelope-associated LDs are enriched with both CEs and TAGs, which contradicts the authors' claim that "these results argue that the NE-localization of SOAT1 assists in the generation of CE-rich LDs in the NE." The authors should provide a clearer explanation for this discrepancy.

This is correct: the nuclear shape is indeed altered due to the artificial tethering of SOAT1 to the nuclear envelope. To address whether the increased NE-associated LDs are associated with these changes in nuclear shape, we generated cells in which a catalytically inactive SOAT1 (SOAT1-H460A) is tethered to the NE using the same strategy as for the WT SOAT1. This shows that the nuclear shape is deformed also in such cells (see **Figure 4 for Reviewer** below). Thus, the altered nuclear shape is not related to the increased activity of SOAT1/increased NE-LDs but rather to the tethering.

Figure 4 for Reviewer. Roundness of nuclei in SOAT NE-trap cells. (a) Mean and standard error of nuclear roundness ($4\pi * area/perimeter^2$) in wild-type, SOAT1 NE-trapped and SOAT1-H460A NE-trapped (inactive SOAT mutant) cells in complete medium or starvation in LPDS. Cell numbers: 3096, 3589, 4788, 2800, 2846, 3271 from left to right. (b) Exemplary images of DAPI staining for cells in complete medium. Scale-bar: 25 μ m.

The Reviewer also mentions that there is a discrepancy between the claim of SOAT1 NE-localization increasing the generation of CE-rich LDs in the NE and the fact that NE-LDs contain both CE and TAG. Indeed, the NE-LDs in this setting do contain some TAGs as well but they are enriched in CEs rather than TAGs and such LDs are more numerous in the setting where SOAT1 is tethered to the NE. We noticed that the image panels in **Fig 5D-E** were perhaps not the best examples and have now replaced them with more representative ones. We also realized that the quantification shown was not explicit and have now improved the accuracy of presentation in the quantification of the relative CE-enrichment in NE-LDs (revised **Fig 5G-H**).

#3/5) To support the conclusion that SOAT1 inactivation or overexpression affects NF- κ B nuclear translocation, the authors should present data for the entire time course (e.g., over 30 minutes), as shown in Figure 6F. Comparing the nuclear translocation at only the 20-minute mark is not sufficient, especially since the observed differences are quite subtle.

Indeed, the differences are rather subtle but systematic. The process is asynchronous at the individual cell level and a high-content imaging approach is needed for quantification. We initially chose to show the 20-minute time point only, as it is around this time when the clearest shift in p65 (NF- κ B) localization can be observed using immunostaining as a readout. As requested, we have now included the nuclear translocation data for the entire time course in the revised **Fig 7D, F** and **Fig 8G**.

#3/6). To clarify the relationship between SOAT1's role in regulating nuclear envelope-associated LDs and NF- κ B nuclear translocation, the authors should not limit their

analysis to SOAT1 overexpression. They should also assess the effects of SOAT1 tethering to the nucleus. Does SOAT1 tethering have a stronger effect on promoting NF-kB nuclear translocation?

We have now also analyzed p65 nuclear translocation in cells where overexpressed SOAT1 is tethered to the nucleus. This did not further increase p65 nuclear translocation (but rather inhibited it compared to SOAT1 overexpression without tethering). As discussed above in our response to point #3/4, the overall nuclear shape is altered in these cells in a SOAT1 activity independent manner. Although the artificial trapping of SOAT1 can be used to demonstrate that the subcellular localization of the enzyme contributes to the subcellular localization of CE-rich LDs, the artificial trapping system most likely compromises the functionality of the nuclear envelope and therefore is not well suited for investigating nuclear translocation kinetics. Instead, to further clarify the relationship between SOAT1's role in regulating NE-associated LDs and NF-kB nuclear translocation, we strengthened the data on the effects of endogenous SOAT1 inhibition in the formation of NE-LDs (new **Fig EV4A-C** and **EV4E-G**), nuclear pore clustering (new **Fig EV5E-F**) and NF-kB nuclear translocation (revised **Fig 7D** and **Fig 8G**).

#3/7). The authors should include SBF-SEM data for control cells to allow for direct comparison and validation of the findings.

As requested, we have now included SBF-SEM data for control cells. These data are provided in **Appendix Fig S1**.

Minor Points:

#3/8) In the sentence "We next want to understand if this acute cholesterol redistribution...", what evidence supports the claim that this redistribution is acute? The authors should provide more information to justify this characterization.

We used the wording acute cholesterol redistribution to refer to the TNF α induced formation of CE-rich NE-LDs. We have now reformulated the sentence to clarify this point (p. 9).

Dear Elina,

We have now received re-review reports from three referees, which I have included below. As you will see, you have addressed their concerns satisfactorily; however, I would like you to consider incorporating the concerns of referee 2 regarding the TNF α stimulation experiments, and include the control shown in Figure 4 for reviewer (in the point-by-point response) in the manuscript appendix as requested by referee 3. Before I can finally accept the manuscript, there are some remaining editorial points which need to be addressed. In this regard would you please:

- label the corresponding authors on page 1 and supply email addresses, provide an OrcID for Dr Vanharanta,
- acknowledge the following funding in our online submission system: Euro-Biolmaging Finnish Advanced Microscopy Node, Biocenter Finland; Sigrid Juselius Foundation and Jane and Aatos Erkko Foundation,
- rename the 'Conflict of Interests Statement' the "Disclosure and competing interests statement",
- remove the AC/CrediT section from the text,
- save the Appendix file in PDF format; the title page should contain "Appendix for Nuclear envelope associated lipid droplets are enriched in cholesteryl esters and increase during inflammatory signaling", include a table of contents with the page numbers for the listed items using the nomenclature 'Appendix Figure Sx' and 'Appendix Table Sx' throughout manuscript and Appendix PDF,
- include a Tools and Reagents table,
- provide p values in the legend of figure 2D,
- indicate the statistical test used for data analysis in the legend of figures 6D and F,
- define box plots in terms of minima, maxima, centre, bounds of box and whiskers, and percentile in the legends of figures 3E, 5F, G; 6C, F; 8B, EV4 C, G; EV5 F,
- define 'n' in the legends of figures EV1, C, F; EV4 B, F, C, G, and
- define error bars in the legends of figures 1A, H; 2B, 5A, C, H, 6D, 8C, E, G, H; EV1 F, EV4B, F.

We include a synopsis of the paper (see <http://emboj.embopress.org/>). Please provide me with a general summary image, a two sentence statement and 3-5 bullet points that capture the key findings of the paper.

I am looking forward to receiving your revised manuscript.

EMBO Press is an editorially independent publishing platform for the development of EMBO scientific publications.

Best wishes,

William

William Teale, PhD
Editor
The EMBO Journal
w.teale@embojournal.org

- a point-by-point response to the referees' comments, with a detailed description of the changes made (as a word file).
 - a word file of the manuscript text.
 - individual production quality figure files (one file per figure)
 - a complete author checklist, which you can download from our author guidelines (<https://www.embopress.org/page/journal/14602075/authorguide>).
 - Expanded View files (replacing Supplementary Information)
- Please see out instructions to authors

<https://www.embopress.org/page/journal/14602075/authorguide#expandedview>
- a Reagents and Tools Table as part of the Methods section, which can be downloaded from our author guidelines
(<https://www.embopress.org/page/journal/14602075/authorguide#structuredmethods>)

We realize that it is difficult to revise to a specific deadline. In the interest of protecting the conceptual advance provided by the work, we recommend a revision within 3 months (20th May 2025). Please discuss the revision progress ahead of this time with the editor if you require more time to complete the revisions. Use the link below to submit your revision:

Referee #1:

The authors have satisfactorily addressed my concerns and strengthened the manuscript with additional studies, analyses, and discussion. Interesting manuscript - congrats.

Referee #2:

This revised version of the manuscript has improved. The text is clearer and some of the additional experiments (SOAT inhibition in Figures 7E-G and EV4) provide some new support for the model. The localization of SOAT1-SNAP is also a good addition however the data presented in Figure 6E-G isn't very compelling. The TNF alpha induced changes in SOAT1 distribution aren't very compelling and it is unclear how the quantification shown in panel 6F was carried out. The enrichment of CE in NE-LDs after addition of TNF-alpha appears underwhelming (Figure 6C). Perhaps it would be justified to tone down some of the claims (such as "these results provide evidence that TNF α induces a shift in SOAT1 distribution closer to the NE, where the enzyme induces the formation of CE-rich NE-LDs" and also in the abstract).

Referee #3:

The authors have thoroughly addressed my comments. The only request left is to add "Figure 4 for Reviewer" to the manuscript and explain that NE tethering affects nuclear size.

EMBOJ-2024-118923R

Szkalitsy, Vanharanta et al. Nuclear envelope associated lipid droplets are enriched in cholesteryl esters and increase during inflammatory signaling

Point by point responses to the referee comments and editorial requests

Referee #1:

The authors have satisfactorily addressed my concerns and strengthened the manuscript with additional studies, analyses, and discussion. Interesting manuscript - congrats.

Thank you very much for the positive evaluation, your kind words and for your valuable comments that helped to improve the manuscript.

Referee #2:

This revised version of the manuscript has improved. The text is clearer and some of the additional experiments (SOAT inhibition in Figures 7E-G and EV4) provide some new support for the model. The localization of SOAT1-SNAP is also a good addition however the data presented in Figure 6E-G isn't very compelling. The TNF alpha induced changes in SOAT1 distribution aren't very compelling and it is unclear how the quantification shown in panel 6F was carried out. The enrichment of CE in NE-LDs after addition of TNF-alpha appears underwhelming (Figure 6C). Perhaps it would be justified to tone down some of the claims (such as "these results provide evidence that TNF α induces a shift in SOAT1 distribution closer to the NE, where the enzyme induces the formation of CE-rich NE-LDs" and also in the abstract).

Thank you for your supportive comments. We have now improved the description of the nuclear proximity readout and generated an additional explanatory figure to explain it better (**Appendix Fig S4**). Indeed, the extent of SOAT1 relocalization to the perinuclear area is moderate but very consistent, as well as the NE-LD increase upon TNF α treatment. As requested, we have toned down the related sentences in the abstract and in the main text.

Referee #3:

The authors have thoroughly addressed my comments. The only request left is to add "Figure 4 for Reviewer" to the manuscript and explain that NE tethering affects nuclear size.

Thank you very much for the positive feedback. We have included the requested figure in the supplementary materials (**Appendix Fig S2**).

Editorial requests:

- label the corresponding authors on page 1 and supply email addresses, provide an OrcID for Dr Vanharanta,

We marked the corresponding author on page 1 and provided her e-mail address. Lauri Vanharanta's ORCID is: 0000-0002-7902-0009 (also provided online).

- acknowledge the following funding in our online submission system: Euro-Biolmaging Finnish Advanced Microscopy Node, Biocenter Finland; Sigrid Juselius Foundation and Jane and Aatos Erkko Foundation,

We acknowledged the requested funding agencies in the on-line system.

- rename the 'Conflict of Interests Statement' the "Disclosure and competing interests statement",

The Conflict of Interests Statement was renamed as requested.

- remove the AC/CrediT section from the text,

We removed the AC/CrediT section from the text.

- save the Appendix file in PDF format; the title page should contain "Appendix for Nuclear envelope associated lipid droplets are enriched in cholesteryl esters and increase during inflammatory signaling", include a table of contents with the page numbers for the listed items using the nomenclature 'Appendix Figure Sx' and 'Appendix Table Sx' throughout manuscript and Appendix PDF,

We prepared the Appendix file in pdf format as requested.

- include a Tools and Reagents table,

We included a Tools and Reagents table in the beginning of the Methods.

- provide p values in the legend of figure 2D,

We included exact p-values in the figure panel for Fig 2D to be consistent with showing p-values in the figure rather than in the legend wherever this is feasible.

- indicate the statistical test used for data analysis in the legend of figures 6D and F,

We marked the used statistical test in Figure 6D.

- define box plots in terms of minima, maxima, centre, bounds of box and whiskers, and percentile in the legends of figures 3E, 5F, G; 6C, F; 8B, EV4 C, G; EV5 F,

To avoid excessively long figure legends, the exact boxplot specifications have been provided in the *Statistics and software* section. A brief reference to this has been added to the legends of all respective panels.

- define 'n' in the legends of figures EV1, C, F; EV4 B, F, C, G, and

We provided 'n' numbers for the requested panels, for each panel jointly at the end of the whole figure legend to be consistent across the figures.

- define error bars in the legends of figures 1A, H; 2B, 5A, C, H, 6D, 8C, E, G, H; EV1 F, EV4B, F.

We specified the error bars in the requested legends.

Dear Elina,

I am pleased to inform you that your manuscript has been accepted for publication in the EMBO Journal.

Congratulations to you and all involved!

Best wishes,

William

William Teale, PhD
Editor
The EMBO Journal
w.teale@embojournal.org
